# Adaptive landscapes unveil the complex evolutionary path from sprawling to upright forelimb function and posture in mammals

Robert J. Brocklehurst [1¤]*, Magdalen Mercado[1,2], Kenneth D. Angielczyk[2,3], Stephanie E. Pierce [1]*

**1** Museum of Comparative Zoology and Department of Organismic and Evolutionary Biology, Harvard University, Cambridge, Massachusetts, United States of America, **2** Committee on Evolutionary Biology, University of Chicago, Chicago, Illinois, United States of America, **3** Negaunee Integrative Research Center, Field Museum of Natural History, Chicago, Illinois, United States of America

¤ Current address: Department of Biological Sciences, University of Massachusetts Lowell, Lowell, Massachusetts, United States of America

* rbrocklehurst@fas.harvard.edu (RJB); spierce@oeb.harvard.edu (SEP)

## Abstract

The 'sprawling-parasagittal' postural transition is a key part of mammalian evolution, associated with sweeping reorganization of the postcranial skeleton in mammals compared to their forebears, the non-mammalian synapsids. However, disputes over forelimb function in fossil synapsids render the precise nature of the 'sprawling-parasagittal' transition controversial. We shed new light on the origins of mammalian posture, using evolutionary adaptive landscapes to integrate 3D humerus shape and functional performance data across a taxonomically comprehensive sample of fossil synapsids and extant comparators. We find that the earliest pelycosaur-grade synapsids had a unique mode of sprawling, intermediate between extant reptiles and monotremes. Subsequent evolution of synapsid humerus form and functional traits showed little evidence of a direct progression from sprawling pelycosaurs to parasagittal mammals. Instead, posture was evolutionarily labile, and the ecological diversification of successive synapsid radiations was accompanied by variation in humerus morphofunctional traits. Further, synapsids frequently evolve toward parasagittal postures, diverging from the reconstructed optimal evolutionary path; the optimal path only aligns with becoming increasingly mammalian in derived cynodonts. We find the earliest support for habitual parasagittal postures in stem therians, implying that synapsids evolved and radiated with distinct forelimb trait combinations for most of their recorded history.

## Introduction

The evolution of mammals is an iconic transition in the history of life that involved the profound modification of key body systems—feeding [1], hearing [2], integument [3],

**Data availability statement:** All data needed to replicate the analyses (landmark coordinates and functional metrics for each specimen, as well as time-dated supertrees) are available in the Supplement. All code needed to analyze the data is available in the supplemental .zip file S1 Data and is based on existing R packages. For replicating raw data collection, custom code for placing landmarks based on the existing R package Morphomap is available as part of S1 Data. 3D models of fossil humeri have been archived with the museum collections which house the original specimens and are available either from those museums (see S8 Table for contacts) or through Morphosource (https://www.morphosource.org/projects/000609110?locale=en).

**Funding:** Funding was provided by the US National Science Foundation (grant DEB1754459 to SEP and grant DEB1754502 to KDA), and by the Harvard Museum of Comparative Zoology (Grant-In-Aid of Undergraduate Research to MM). Funding for publication costs provided by a grant from the Wetmore Colles Fund. The funders had no role in study design, data collection and analysis, decision to publish, or preparation of the manuscript.

**Competing interests:** The authors have declared that no competing interests exist.

**Abbreviations:** bgPCA, between-groups principal components analysis; BM, Brownian motion; NMS, non-mammalian synapsids; OU, Ornstein–Uhlenbeck; PCA, principal components analysis; RO, optimal ranking; RS, suboptimal ranking.

and metabolic physiology [4]. The exceptionally rich fossil record of mammals and their stem lineage, the non-mammalian synapsids (NMS), documents the assembly of these traits in great detail over some 300 million years [5,6]. Of particular interest is the dramatic reorganization of the ancestral synapsid postcranial musculoskeletal system, including regionalization of the backbone; simplification of the shoulder girdle; evolution of novel joint types (e.g., ball-and-socket shoulder, trochlear elbow); major restructuring of the limb musculature; and reorientation of the limbs from a horizontal to vertical plane [7–11]. These broad-scale anatomical transformations are intimately associated with a functional shift in limb posture and locomotion, from sprawling pelycosaur-grade synapsids to parasagittal therian mammals.

Although the synapsid 'sprawling-parasagittal' transition was a key event in mammalian evolution, its precise nature remains controversial [5,12–15]. Historical studies portrayed synapsid evolution as a stepwise, linear progression toward the therian condition, but often disagreed over when and how major anatomical changes translated into functional or postural change [5,12,13]. These issues primarily arose because previous authors focused on "exemplar" fossil taxa at different key nodes in the synapsid phylogeny and were restricted to qualitative functional interpretations of bony morphology. Recently, more taxonomically comprehensive morphometric work on the synapsid forelimb suggests a pattern of successive evolutionary radiations, with major synapsid groups exploring distinct morphologies and presumed functions [14,16]. However, to date, few studies on synapsid postcrania have incorporated explicit links between form and function into an analytical framework [17,18], a crucial step in characterizing the origins of therian-like limb posture and parasagittal locomotion.

Here, we use the forelimb, particularly the humerus, as a lens to study postural evolution in synapsids. Forelimb modifications were key to the evolutionary success of synapsids, including mammals [7,8,14–16], and the humerus provides an important window into forelimb function and posture: it is the primary articulation point between the forelimb and body, anchors the major muscle groups that drive locomotion, and its arc of motion directly impacts limb movements [10,19–21]. Further, isolated humeri preserve well in the fossil record and so allow for more extensive taxonomic sampling to produce a comprehensive evolutionary viewpoint on the 'sprawling-parasagittal' transition. We analyze humerus shape and functional traits across >200 extant and extinct taxa using the integrative analytical framework of evolutionary adaptive landscapes [18,19,22]. This framework links form to functional performance across multiple traits, permitting inference between morphology and higher-level functions such as limb posture or locomotion [19,23], and accurately captures the evolution of functional tradeoffs [24]. We extend this synthesis with Pareto optimality analyses [25] to examine how synapsids transitioned between performance peaks and adaptive optima throughout their evolution and across the 'sprawling-parasagittal' postural shift.

Given the differing biomechanical requirements of sprawling versus parasagittal locomotion [15,26,27] and the historical perception of a progression toward therian parasagittal posture [5,12,13], we tested the following hypotheses: (1) Humeri from

sprawling and parasagittal extant taxa occupy different adaptive optima, each maximizing traits most relevant to their specific limb posture and gait; (2) "Pelycosaurs", the earliest-diverging NMS, share an adaptive optimum with extant sprawling taxa; (3) Increasingly derived grades of NMS shift their adaptive landscapes toward extant therians as they achieve more parasagittal postures; and (4) Evolutionary changes in NMS morphology, functional traits, and posture follow an optimal pathway from sprawling to parasagittal. Our data reveal morphological and functional trait similarities between NMS and extant comparative taxa, but also some key differences. We find that certain performance traits strongly correlate with posture, whereas others relate to different aspects of forelimb function. Recovered patterns show that postural evolution within Synapsida was complex, and the 'sprawling-parasagittal' transition was characterized by homoplasy and functional variation within individual synapsid clades, indicative of multiple adaptive radiations of NMS. Morphofunctional traits consistent with fully parasagittal posture evolved late in the evolutionary lineage of mammals, and so for the majority of synapsid history, the forelimbs were characterized by patterns of variation in form, function, and posture that are distinct from what we see in therians today.

## Results

### Humerus shape variation

To capture morphological evolution of the humerus throughout synapsid evolution, we measured humeri from 70 fossil taxa and compared them to 141 extant quadrupedal tetrapods including amphibians, reptiles, and mammals (see S1 Table and S1 Fig). Shape variation was quantified using a novel, homology-free pseudo-landmarking approach to generate 3D coordinates along the surface of each bone (see Methods and S2 Fig). Procrustes aligned landmarks were ordinated using between-groups principal components analysis (bgPCA) to differentiate postural groups (sprawling versus parasagittal versus unknown for fossils) and produce a morphospace (Fig 1). Regarding our first hypothesis, humeri of major extant groups fall in distinct regions of morphospace (Procrustes ANOVA and pairwise comparisons, $p < 0.05$, see S2 Table), but the main axis of shape variation (bgPC1) does not separate them based on posture. Instead, bgPC1 differentiates the relatively gracile humeri of parasagittal therian mammals and sprawling "herptiles" (non-avian reptiles plus amphibians), from the robust humeri of sprawling monotremes and moles (Fig 1A and 1B). Therians separate from herptiles along the second axis of variation (bgPC2), which reflects differences in the offset between the proximal and distal ends of the humerus, curvature of the humeral shaft and relative width of the proximal versus distal ends of the humeral epiphyses (Fig 1A and 1B).

Pelycosaurs, the earliest grade of NMS, occupy the region of morphospace between herptiles and monotremes—contra hypothesis 2—while overlapping neither, consistent with their "basal" phylogenetic position, and a unique sprawling posture [13,28] (significant pairwise differences between means, pelycosaurs versus herptile $p < 0.05$, pelycosaurs versus monotremes $p < 0.05$, see S2 Table). Other non-synapsid fossil taxa also plot in this part of morphospace (e.g., *Eryops*, *Orobates*, *Seymouria* [20,22]), indicating pelycosaur humeri do not deviate strongly from a general early crown-tetrapod condition. However, more derived grades of NMS—therapsids and cynodonts—diversify into much larger regions of morphospace and exhibit greater variation in humerus shape (Figs 1A, 1C, and S3).

In therapsids, this variation is partitioned across different subclades. Anomodonts and dinocephalians generally plot with pelycosaurs or closer to monotremes. The biarmosuchian *Hipposaurus*, our earliest-branching therapsid, plots within herptiles, gorgonopsians plot on the outer edges of the therian and herptile spaces, and therocephalians plot close-to or within the therian cluster (Fig 1C). Basal cynodonts generally occupy the central region of morphospace, as do the two more derived eucynodont subclades, cynognathians and probainognathians, but in both subclades some species independently move into the therian region of morphospace (see S3 Fig). Some mammaliaforms—*Eozostrodon* and *Megazostrodon* [29]—fall within therian morphospace, but others—*Borealestes* [30]—do not (Figs 1C and S3). The two stem therians included here fall well within the range of extant, crown-therian morphologies (Figs 1A and S3). While the overall mean humerus shape of each successive NMS radiation—pelycosaurs, therapsids, and cynodonts—gets closer to

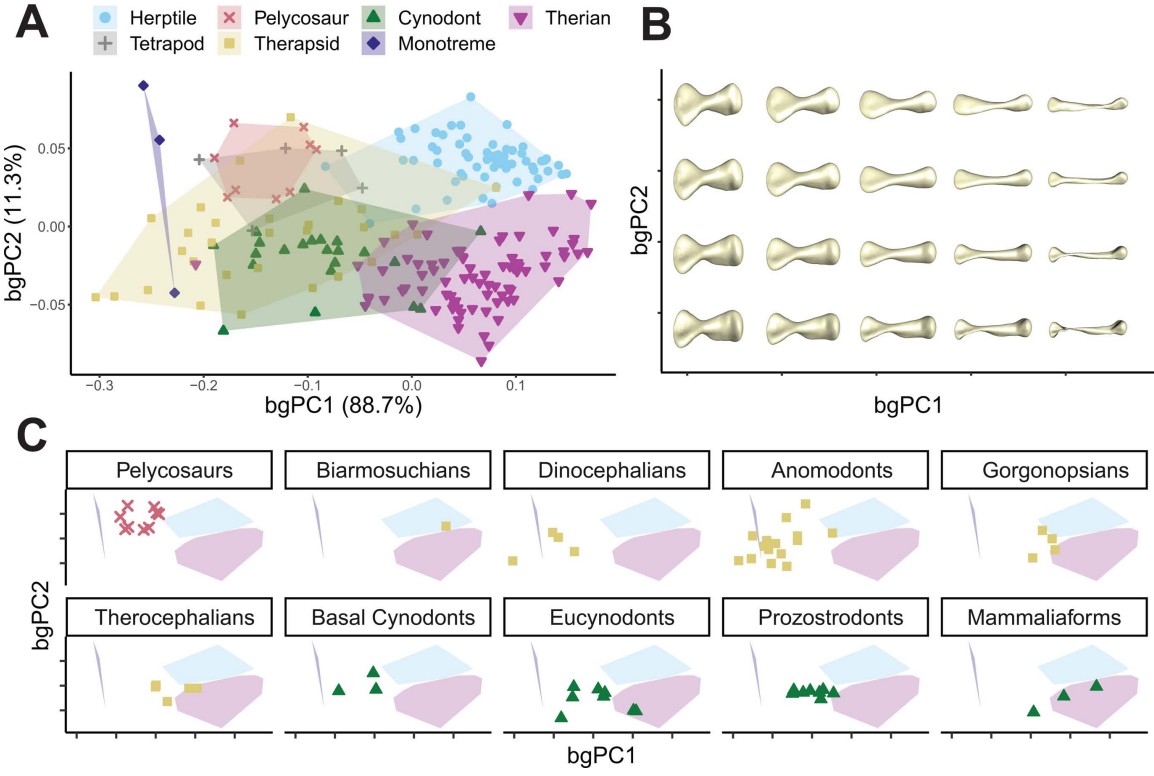

**Fig 1. Humerus morphological variation. (A)** Humerus between-groups principal components analysis (PCA) morphospace showing major axes of shape variation. **(B)** Warps illustrate representative humerus morphologies at each point in morphospace. **(C)** Subplots show positions of non-mammalian synapsid subgroups, relative to the convex hulls of the three extant groups. Axes are labeled with the percentage of between-groups variance explained. See S11 Fig for a comparison between PCA and bgPCA results. The data underlying this figure can be found in S1 Table and S1 Data.

the mean therian shape (see pairwise distances in S2 Table), as outlined by hypothesis 3, there is considerable variation around these mean shape values.

## Functional traits, performance surfaces, and trait optimization

Function was inferred by measuring seven osteological proxies—bone length, radius of gyration (an important inertial property), torsion, bending strength, and muscle force and speed leverage—for each humerus in our sample (see Methods and S4 Fig). Muscle leverage was separated into two components; 'swing' refers to motion of the limb through a horizontal or vertical arc during locomotion, and 'spin' refers to long-axis rotation of the limb [15]. We interpolated the measured functional traits across the morphospace [31], to produce seven distinct performance surfaces that illustrate the co-variation of humerus shape with each functional trait (Fig 2A). Performance surfaces were combined to produce adaptive landscapes for each species in the dataset using combinatorial optimization, weighting functional traits to maximize each species' height on their resulting landscape [18,19,22]. Combinations of trait weights that produced optimal landscapes were then compared across taxa and along the synapsid phylogeny using ancestral state reconstructions (Fig 2B and 2C). Evolutionary shifts in functional trait weighting were identified using the SURFACE algorithm [32], and specific a priori hypotheses based on previous studies of synapsid locomotor evolution [5,12,13] (see S5 Fig) were tested against each other using mvMORPH [33] (see Methods).

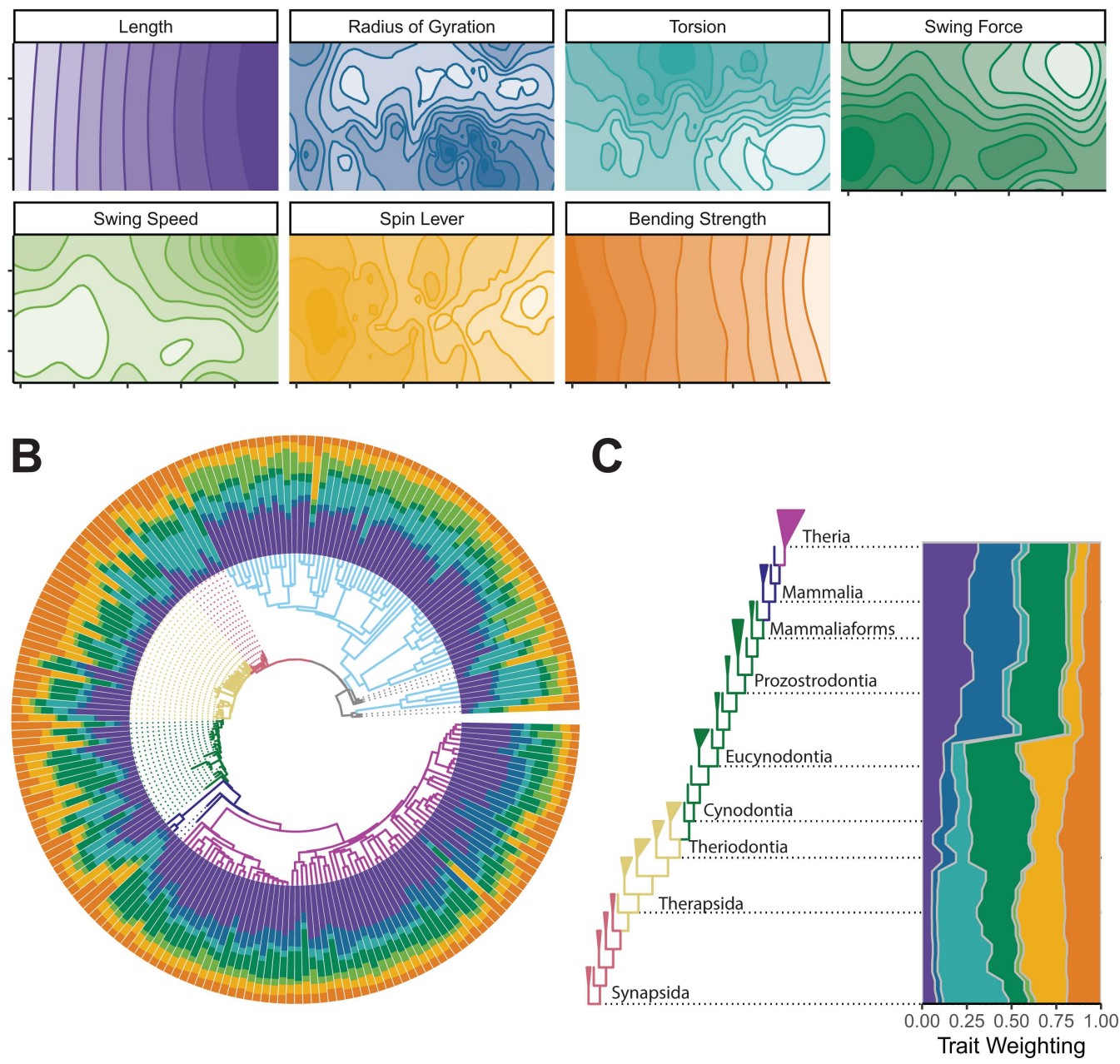

**Fig 2. Functional trait variation. (A)** Performance surfaces of seven measured functional traits. **(B)** Trait weightings of adaptive landscapes calculated for individual species plotted onto the phylogeny. **(C)** Weightings of adaptive landscapes reconstructed at ancestral nodes plotted against a simplified phylogeny of non-mammalian synapsids. The data underlying this figure can be found in S1 Table and S1 Data.

Species-level analysis of trait weights shows considerable variation across groups (RRPP MANOVA, $p < 0.05$, $r^2 = 0.39$, see S3 Table), but also within major groups, especially within NMS (Figs 2B and S6). Centroid size also showed a statistically significant, but smaller, effect on trait weightings (RRPP MANOVA, $p < 0.05$, $r^2 = 0.11$, see S3 Table), and

is likely important in structuring within-group variation. Examining adaptive landscapes based on ancestral-state reconstructions of humerus shape along the backbone of the synapsid phylogeny (Fig 2C) reveals the relative importance of individual traits through synapsid evolution, semi-independent of clade- or taxon-specific patterns. The reconstructed ancestral landscape for the Synapsida node heavily optimizes humeral torsion, as well as 'spin' muscle leverage, and this combination remains stable throughout early synapsid evolution. The landscape changes at the base of Theriodontia (gorgonopsians, therocephalians, and cynodonts), and throughout Cynodontia, with an increase in optimization for humeral length, 'swing' force leverage and a decrease in weighting for torsion (Fig 2C). Notable changes to the reconstructed landscapes occur prior to the origins of prozostrodontian cynodonts and Mammaliaformes: increased optimization for humeral length, radius of gyration, and 'swing' force leverage, with less of an emphasis on humeral torsion and 'spin' leverage (Fig 2C).

In addition to individual adaptive landscapes for each specimen, we calculated adaptive landscapes for major taxonomic groups within our dataset; here, the group mean is maximized on the resulting landscape. Extant species optimize different combinations of functional traits (Figs 2, 3, and S6), in support of hypothesis 1. Herptiles strongly emphasize humerus length and humeral torsion (Fig 3), traits associated with increasing stride length in sprawling tetrapods [34,35], as well as high-velocity advantage for humeral 'swing' (Fig 3), resulting in relatively fast limb movements and greater muscle working range [15]. Monotremes show high weighting for humeral torsion, 'spin' muscle leverage, and bending strength (Fig 3), reflective of both their sprawling posture and semi-fossorial lifestyle [15,36–38]. Therians optimize humerus length, radius of gyration, and 'swing' muscle force leverage for rotating the limb through an arc (Fig 3). Longer length and a shorter radius of gyration are associated with more efficient locomotion [39,40], whereas increasing muscle leverage for planar rotation would result in more powerful movements of the limb in the parasagittal plane [15,41].

Individual NMS vary substantially in how different functional traits are optimized on their landscapes (Figs 2B, 3, S6, and S7). Pelycosaurs primarily emphasize humeral torsion, consistent with their reconstructed sprawling posture [42,43]. Other traits—'spin' leverage, 'swing' force, and humeral strength—also contribute to the pelycosaur landscape with varying importance across taxa (Figs 3 and S7). Pelycosaurs differ significantly in terms of trait weighting from herptiles, contra hypothesis 2, but not from monotremes (see S3 Table). Therapsids differ in functional trait optimization across subclades (Figs 2B, 3, S6, and S7). The biarmosuchian *Hipposaurus* emphasizes torsion and humeral length, similar to herptiles. Dinocephalians and anomodonts predominantly optimize strength and muscle force leverages. Gorgonopsians and therocephalians optimize humeral length, but whereas gorgonopsians emphasize 'swing' torque and strength, therocephalians put more importance on humeral torsion (Figs 3 and S7). Cynodonts show substantial variability in trait optimization, but generally emphasize either humerus length, or a combination of strength and muscle force leverages (Figs 2B, 3, S6, and S7). Most cynodonts do not optimize humeral torsion, but there are some exceptions including among derived prozostrodonts and mammaliaform (S6 and S7 Figs). This contrasts with therian trait optimization, as well as that reconstructed along the synapsid backbone—the importance of torsion in these taxa may represent independent instances of trait diversification. As with humerus shape, the mean trait weightings of each successively derived NMS grade become more similar to the therian condition, in support of hypothesis 3. However, within each grade, there is substantial variation in trait optimization, which does not support the idea of a series of stepwise shifts toward therians.

Searching for shifts in how functional trait weightings may have evolved using SURFACE matched those from the ancestral state reconstruction for the backbone of the synapsid phylogeny (Figs 2 and S6). This analysis also recovered additional shifts within synapsid sub-groups, or along individual branches (S6 Fig). Fitting of different evolutionary regimes to the data using mvMORPH found historical hypotheses of morphofunctional shifts within synapsids [5,12,13] are significantly outperformed by more complex models featuring additional optima for subgroups within the major synapsid grades (see S4 Table for detailed model comparisons). These results are more consistent with the view of synapsid evolution as a sequence of radiations, with functional differentiation within and between each major grade.

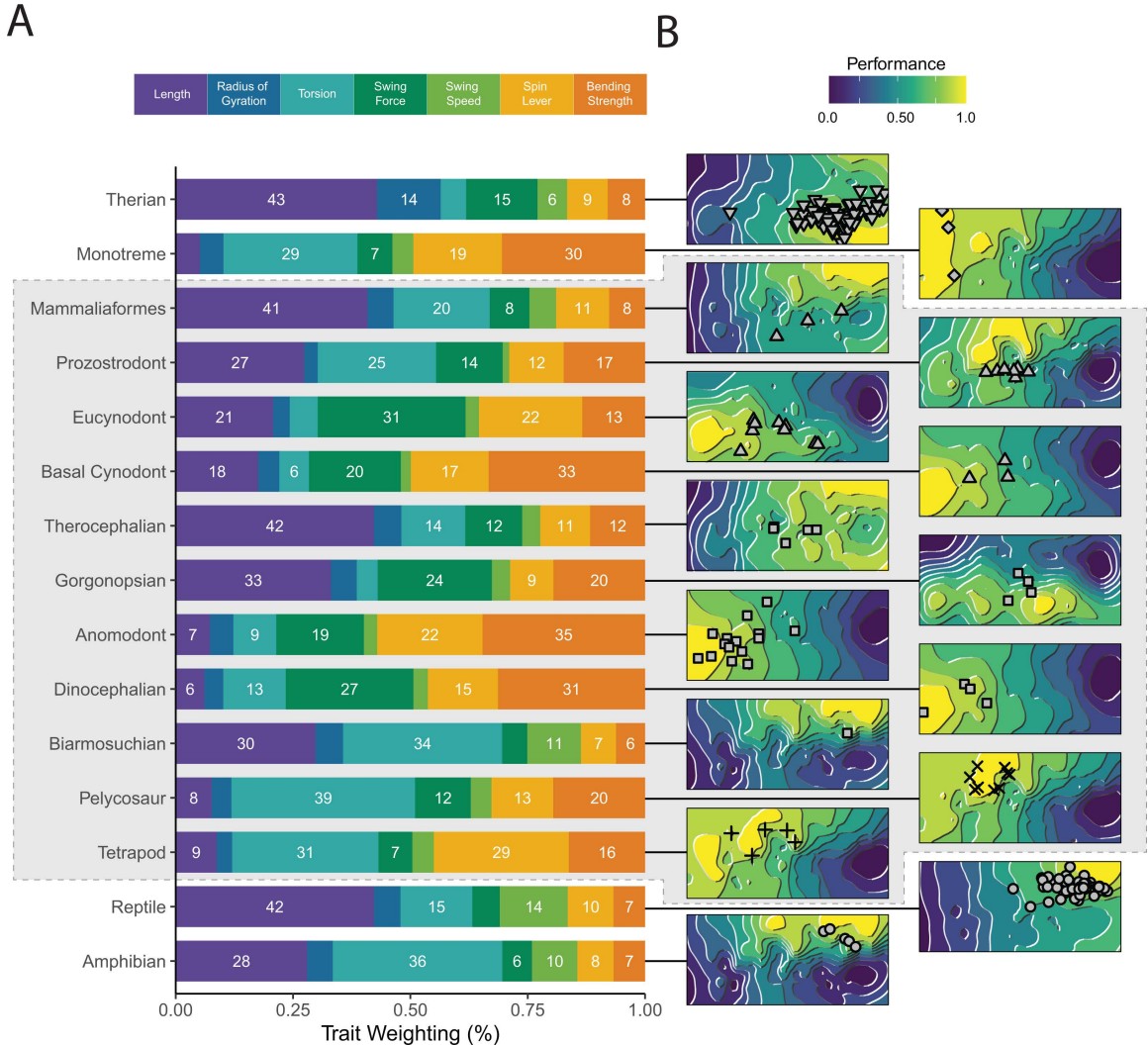

**Fig 3. Adaptive landscapes and underlying trait weights. (A)** Trait weightings of adaptive landscapes calculated for major extant and fossil groups, which maximize the performance of the group mean humerus shape. **(B)** The reconstructed adaptive landscapes for each group, showing performance peaks and valleys across morphospace. Fossil taxa are indicated by the gray outline. The data underlying this figure can be found in S1 Table and S1 Data.

## Transitions from sprawling to parasagittal postures

To further investigate patterns of humeral transformation toward (or away from) the therian condition during the evolution of Synapsida, we created a transitional 'sprawling-parasagittal' landscape. We first created a composite "sprawling" landscape by combining the independently calculated landscapes for different sprawling groups (Fig 4A and see Methods). Then, for each point in morphospace, we subtracted the height on the "sprawling" landscape from the height on the therian (i.e., "parasagittal") landscape (Fig 4B) to create the transitional landscape (Fig 4C). Transitional landscape scores indicate the relative performance of different humerus morphotypes for sprawling versus parasagittal limb functions, given the many-to-one mapping that exists between humeral traits and posture [20,36,44]. We plotted our taxa on this new 'sprawling-parasagittal' landscape (Fig 4C and 4D), determined their adaptive score (height), mapped these scores onto

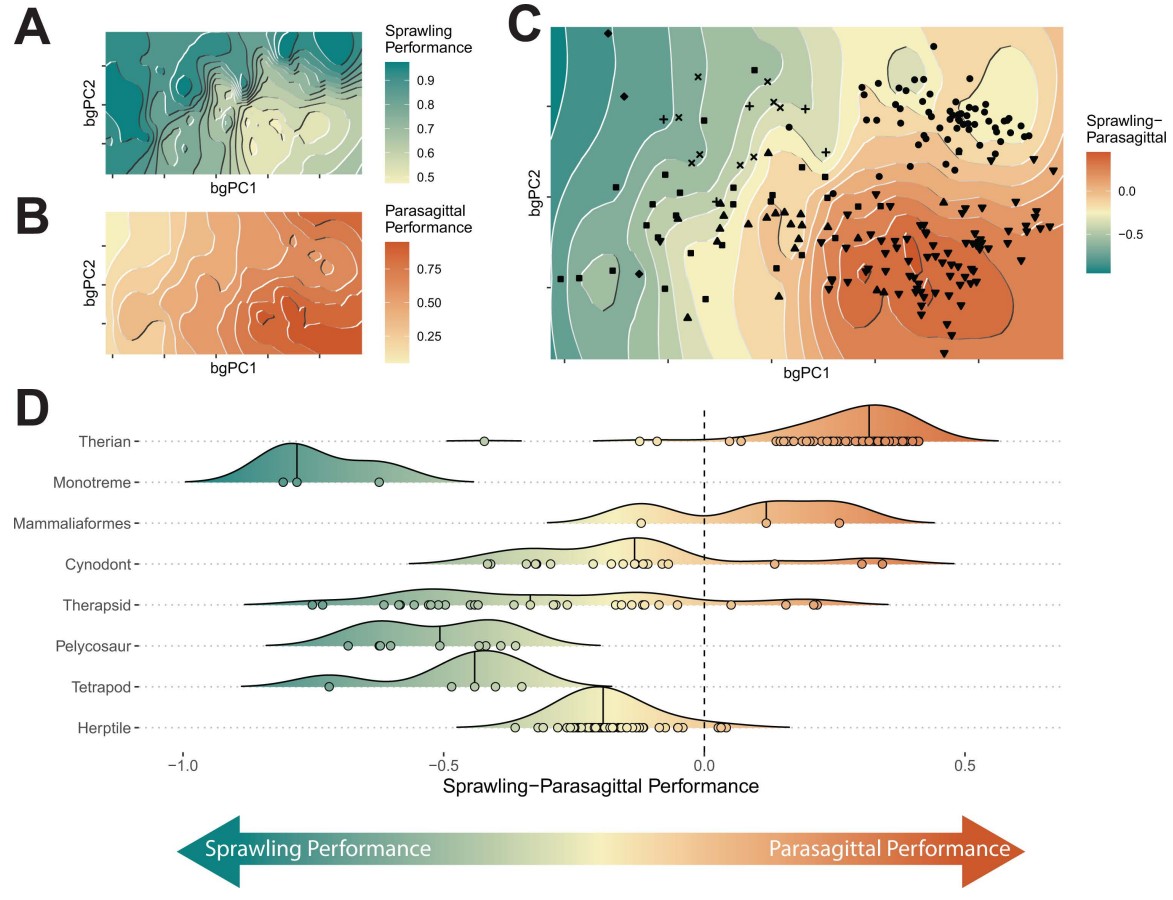

**Fig 4. Transitional 'sprawling-parasagittal' landscape. (A)** Composite sprawling landscape (reptiles, salamanders, monotremes, and non-synapsid fossils), and **(B)** parasagittal (therian) landscape, **(C)** overlaid to create a transitional 'sprawling-parasagittal' landscape. **(D)** Density plots of scores on the transitional 'sprawling-parasagittal' landscape plotted for major extant and fossil groups. Points represent individual specimen scores. The data underlying this figure can be found in S1 Table and S1 Data.

a phylogeny and reconstructed ancestral states using maximum likelihood (Fig 5). As with our functional traits, we used SURFACE and mvMORPH to test for evolutionary shifts both a posteriori and a priori in transitional landscape score within Synapsida and compared models of postural evolution in this group (see Methods).

We find statistically significant differences in postural scores between taxonomic groups with no statistical effect of centroid size (RRPP ANOVA, $p < 0.05$, see S5 Table). As expected, therians had the highest scores, and monotremes had the lowest, indicating trait combinations most and least associated with parasagittal postures respectively (Fig 4C and 4D). Herptiles were intermediate between monotremes and therians, due to sharing morphofunctional traits with both groups (humeral torsion with monotremes, and long humeri with therians), but still had negative scores indicating greater association with sprawling locomotion (Fig 4D). These results are consistent with hypothesis 1. Thus, the transitional landscape helps to conceptualize posture as a continuous variable; sprawling and parasagittal represent extremes, but we also show the great variation present within sprawlers and multiple paths between different postural groups [20,44,45]. Ancestral state reconstruction recovers the ancestor of crown-mammals as closer to therians than monotremes (Fig 5). Consequently, extant sprawling monotremes do not retain a plesiomorphic state for Synapsida but have rather converged on a suite of morphofunctional characters that resemble ancestral, sprawling synapsids. This convergence may have been

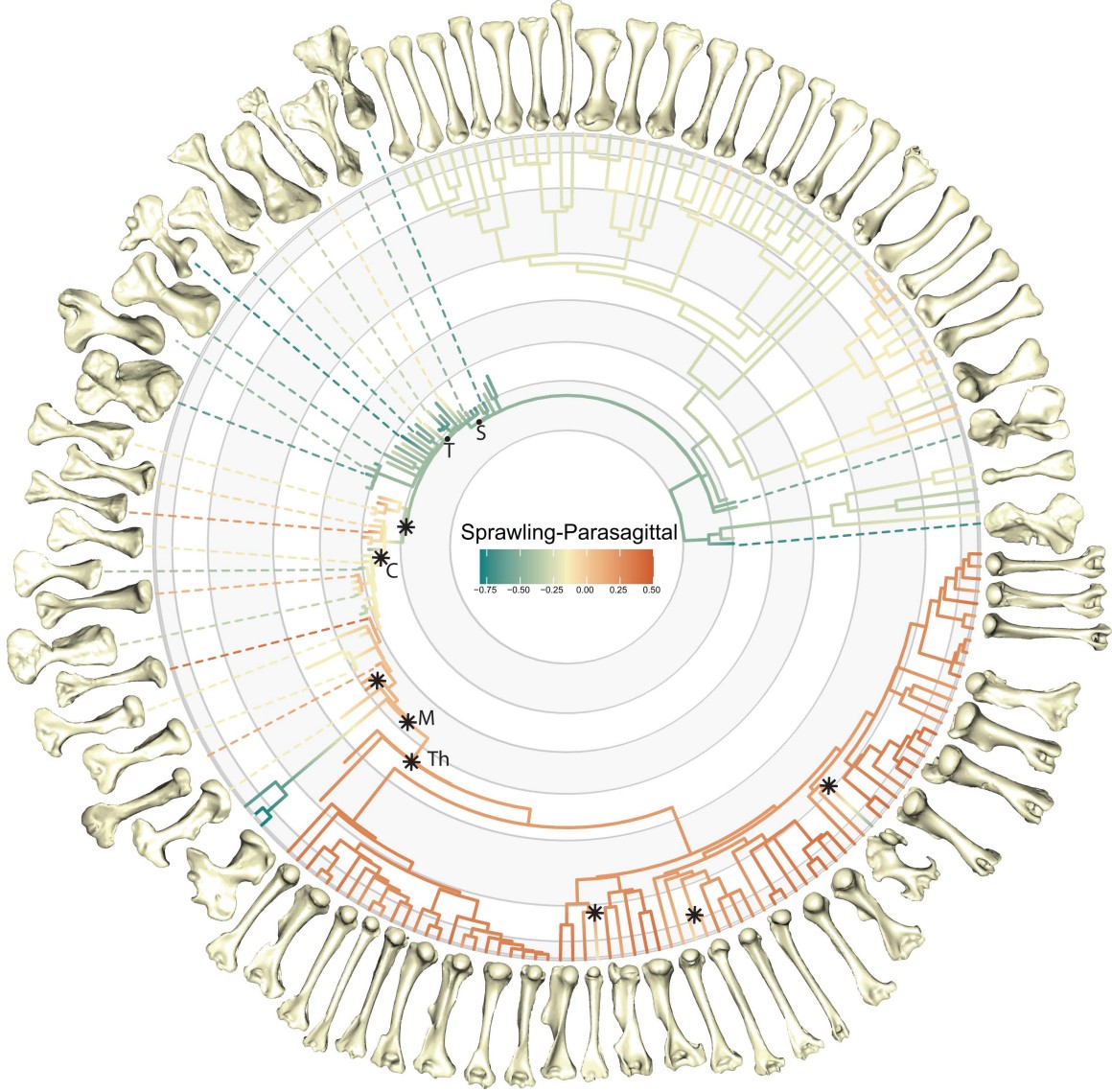

**Fig 5. Evolution of sprawling and parasagittal posture.** Scores on the transitional 'sprawling-parasagittal' landscape are plotted on the phylogeny of all specimens, illustrated with select humeri. Stars indicate shifts in evolutionary regime for Synapsida identified by SURFACE (see Methods). Nodes labeled with letters; S, Synapsida; T, Therapsida; C, Cynodontia; M, Mammalia; Th, Theria. The data underlying this figure can be found in S1 Table and S1 Data.

facilitated by the incomplete acquisition of therian characters in early mammals and their close relatives (e.g., retention of humeral torsion in taxa such as *Brasilodon*, *Morganucodon*, and *Gobiconodon*).

Our analyses reconstruct ancestral humeri at the base of Synapsida with traits consistent with sprawling postures, which persist throughout the pelycosaur and early therapsid parts of the tree (Figs 4D and 5). Pelycosaurs, dino-cephalians, and most anomodonts have humeri with very negative scores on the transitional landscape, providing strong evidence for sprawled postures (Figs 5 and S8). Although our analysis indicates a sprawled ancestor for Therapsida with monotreme-like traits, several early therapsids show less negative scores (e.g., *Hipposaurus*, *Tiarajudens*), more in-line

with modern reptile values, possibly indicative of differences in locomotion (e.g., faster limb movements and different kinematics) (Figs 4D, 5, and S8). We also recovered a shift toward intermediate postural scores in theriodont therapsids: the gorgonopsians and therocephalians (Figs 5 and S9). Although these groups have humeri with distinctly higher transitional landscape scores than earlier synapsids, their scores are generally still negative, like modern reptiles. Another shift on the transitional landscape occurs at the origin of Cynodontia, with most taxa possessing intermediate postural scores (Figs 4D, 5, S8, and S9). However, there is considerable heterogeneity within cynodonts and later mammaliaforms, with taxa evolving humeri with both more negative (e.g., *Chiniquodon*, *Exaeretodon*) and more positive scores (e.g., *Massetognathus*, *Probainognathus*) (Figs 5 and S8). By contrast, within crown mammals, the two stem therians (*Adalatherium* and *Gobiconodon*) both scored positively on the transitional landscape (Fig 5).

In line with the complex series of shifts recovered by SURFACE (Figs 5 and S9), we found the best fitting a priori models were those with additional subgroup optima within the major synapsid grades. By contrast, models with only three regimes—here corresponding to proposed sprawling, intermediate and parasagittal postural groups [5,12,13]—are poor fits to the data (see S6 Table for detailed model comparisons). Once again, we interpret these results as evidence for synapsid evolution as a sequence of radiations, with both functional and postural variation within and between groups.

### Performance trade-offs and optimal evolutionary pathways

To determine whether synapsids followed "optimal" paths through morphospace over the course of their evolution as they transitioned from one adaptive optimum to another, we calculated Pareto optimality landscapes [25,46]. Trade-offs necessarily occur between high fitness on different landscapes due to differences in the underlying functional traits being optimized. Pareto optimality finds points in morphospace whose height on one landscape is maximized, given their height on another landscape [25,47]. By combining adaptive landscapes reconstructed for major nodes in the synapsid phylogeny, we determined if taxa evolve optimally through morphospace from an ancestral peak to that of the next more derived taxonomic group (see Methods). Deviation from high Pareto optimality implies exploration of novel morphologies and functions, and either the presence of new functional drivers distinct from those along the synapsid backbone or the weakening of functional constraints and lowering of restrictions on limb evolution over time [25].

Contrary to expectations that synapsids would follow a consistently optimal path toward a more therian morphology, our results based on reconstructed landscapes for major nodes show statistically significant fluctuations in optimality across the major synapsid groups (with marginally non-significant effects of centroid size; RPPP ANOVA, $p < 0.05$, see S7 Table). This pattern is driven by individual subclades branching off to explore both more and less optimal regions of morphospace (Figs 6 and S10). Pelycosaurs undergo a small radiation but are mostly restricted to regions of high optimality that connect their adaptive peak with that for the reconstructed therapsid ancestor (Fig 6). Therapsids have lower optimality than pelycosaurs, as several clades—biarmosuchians, gorgonopsians, and therocephalians—explore parts of morphospace with lower optimality on the landscape connecting Therapsida with Cynodontia (Figs 6 and S10). Dinocephalians and anomodonts both exhibit greater Pareto optimality on this landscape, clustering around the inferred Therapsida optimum (Figs 6 and S10). Basal cynodonts and eucynodonts generally occupy regions of lower Pareto optimality, but certain taxa evolve toward distinct peaks on the cynodont-prozostrodont landscape, with some species evolving toward the reconstructed ancestral cynodont optimum, associated with more robust humeri, and others evolving toward the prozostrodont optimum, associated with more gracile humeri (Fig 6). Finally, prozostrodont cynodonts and mammaliaforms also occupy sub-optimal regions, although there does seem to be a shift in morphospace occupation toward the optimum reconstructed for ancestral therians (Figs 6 and S10).

## Discussion

The evolution of parasagittal posture in mammals and their ancestors has been studied for over a century, but previous efforts to understand changes to the limbs and locomotion have historically lacked taxonomic scope and an

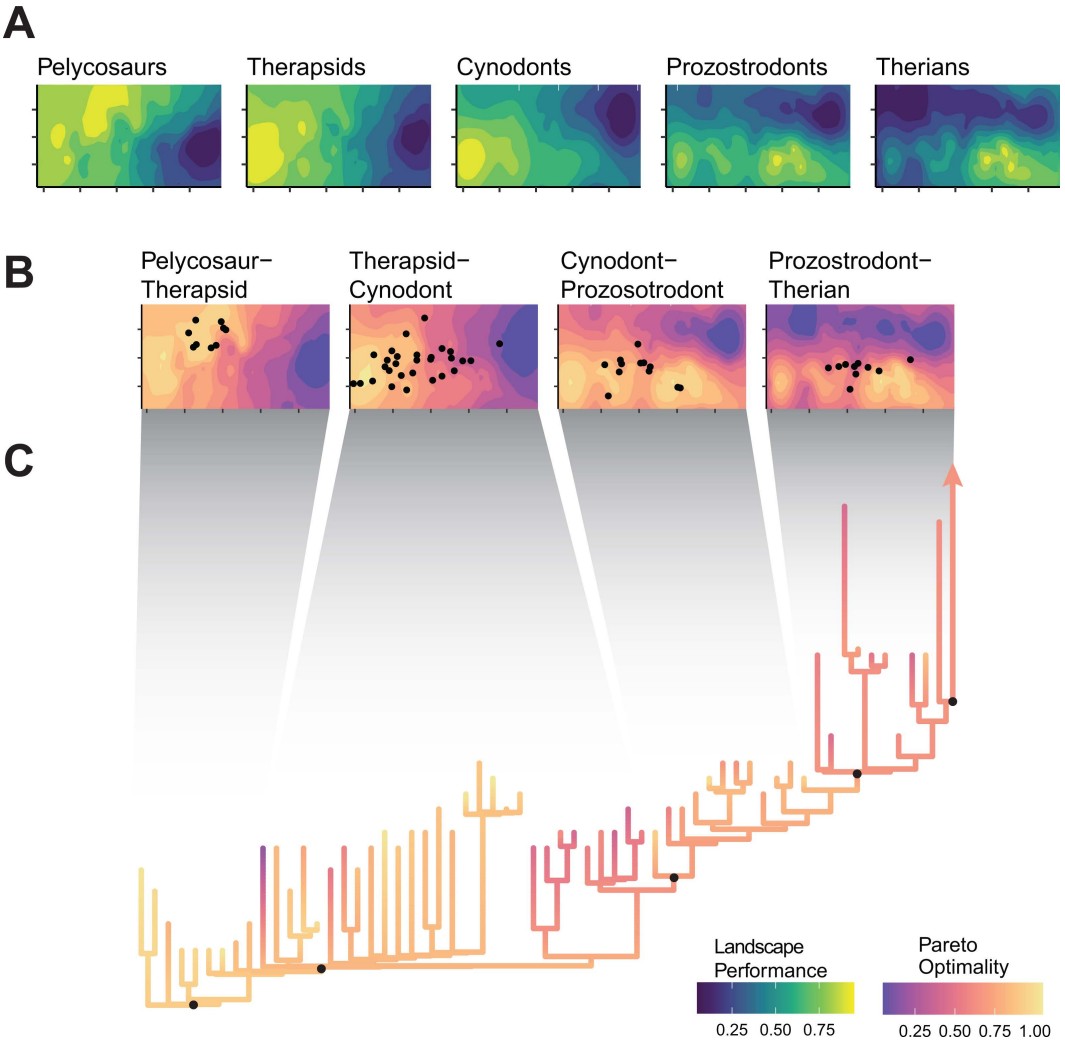

**Fig 6. Pareto optimality across synapsid evolution.** (A) Functional adaptive landscapes reconstructed for major ancestral nodes within synapsid evolution and (B) the Pareto landscapes created by combining these landscapes together. Pareto optimality for a group of taxa is defined based on their ancestral peak, and the peak of the next major node. (C) Pareto optimality plotted on the non-mammalian synapsid phylogeny. The data underlying this figure can be found in S1 Table and S1 Data.

appropriate analytical framework [5,12,13]. Here, we used evolutionary adaptive landscapes [18,22] to predict relationships between humerus morphology, functional traits, and posture to develop a predictive framework to illuminate the 'sprawling-parasagittal' transition in synapsids. Based on differences in locomotor biomechanics, we expected the humeri of extant taxa to group together in morphospace based on posture (Hypothesis 1). Although parasagittal therians group separately from other taxa, sprawling reptiles and amphibians ("herptiles") do not group with sprawling monotremes. Together with therians, herptiles share relatively long, gracile humeri (Figs 1, 3, and S6), features that arose convergently in these groups despite their contrasting habitual postures. Longer humeri are likely advantageous for general terrestrial locomotion regardless of posture, although the precise selective advantages may differ (e.g., speed versus efficiency) [34,39,40]. In contrast to therians, herptiles and monotremes both emphasize humeral torsion, supporting torsion as a strong indicator of sprawling forelimb posture [35,48] (Figs 2–4). However, monotremes and herptiles optimize torsion to

differing degrees and in distinct combinations with other functional traits, reflecting wide variation in sprawling locomotor kinematics (e.g., powerful long-axis rotation versus faster limb retraction) [20,36,37]. Therians are unique in emphasizing radius of gyration (Figs 2 and 3)—due to a combination of a more proximally located center of mass and reorganization of the humeral head moving the center of rotation distally—potentially indicating this trait as a signal of more parasagittal postures. These similarities and differences across diverse extant animals provide interpretive power when reconstructing forelimb function and postural evolution in synapsids.

The earliest-diverging NMS, the pelycosaurs, are traditionally reconstructed with a sprawling posture [42], and thus we hypothesized morphological and functional similarities with extant sprawlers (Hypothesis 2). Yet, our analysis shows pelycosaurs are morphologically distinct from all extant sprawling groups, and instead overlap with extinct non-synapsid tetrapods, indicating that pelycosaur humeri had not diverged from the plesiomorphic crown tetrapod condition (Fig 1). Functionally, pelycosaurs show high optimization for torsion, indicative of sprawling posture, but they optimize this trait to a more extreme degree than either reptiles or monotremes (Figs 3 and S6). Pelycosaurs also emphasize both 'swing' force and 'spin' muscle leverage, which aligns with previous work demonstrating coupled rotations at the screw-shaped glenoid joint during a stride [13,28], as well as humeral strength. Taken together, the emphasis on torsion, strength, and muscle leverage shows pelycosaurs used slow, forceful limb movements, combining humerus long-axis rotation and retraction. This combination contrasts with reptiles, which are adapted for (relative) speed [15], and monotremes, which use almost exclusively long-axis rotation [36,37]. Our analyses thus indicate pelycosaurs possessed distinct humerus morphologies and combinations of functional traits not represented by modern taxa [18], and it was from this unique starting point that mammalian posture ultimately evolved.

Based on historical interpretations of the synapsid fossil record [5,12,13], we expected a trend toward more therian-like morphologies and functional traits in more derived NMS, reflecting more parasagittal postures (Hypothesis 3). Although this general pattern was broadly supported, we also found significant variations on this theme (Figs 1–3). Our transitional 'sprawling-parasagittal' landscape strongly supports monotreme-like sprawling postures in dinocephalian and anomodont therapsids (Figs 4 and 5), with these clades showing increased optimization for strength and muscle force leverages at the expense of humerus length (Figs 2, 3, S6, and S7). This combination of functional traits is potentially related to the acquisition of larger body sizes in both groups [49,50], and fossorial behaviors in smaller dicynodonts [51–53], as seen in modern monotremes and talpid moles [36,37,54]. Similar ecomorphological convergence in the forelimb has been previously noted between fossorial and large-bodied mammals [14]. We argue that the co-occurrence of large body sizes and strong humeri in dinocephalians and anomodonts is indicative of non-parasagittal postures in these groups. Modern large-bodied therians can compensate for size-related bending stresses by changing limb posture, from more crouched to more erect [26]. Sprawling taxa, on the other hand, must accommodate size-related stresses by increasing limb bone robusticity, resulting in "overbuilt" limb bones [27,55].

Other therapsid groups had diverging scores on the 'sprawling-parasagittal' transitional landscape. Biarmosuchians, the earliest branching therapsids and represented here by *Hipposaurus*, experienced a shift on the transitional landscape away from monotremes and toward reptiles (Figs 4, 5, S8, and S9). Specifically, *Hipposaurus* overlaps with reptiles in morphospace and optimizes humeral torsion, length and 'swing' speed (Figs 1 and 3), indicating a reptile-like sprawling posture and kinematics [20]. Although this combination suggests adaptations for relatively fast limb movements, it may not be reflective of the ancestral therapsid condition, due to a persistent ghost-lineage following their divergence from pelycosaurs [56,57]. Later theriodont therapsids, the gorgonopsians and therocephalians, experienced similar but less extreme shifts on the transitional landscape, sitting on the outskirts of the reptile and therian regions of morphospace (Figs 4, 5, S8, and S9). Therocephalians share similar functional traits to *Hipposaurus*, but gorgonopsians combine humerus length with 'swing' force and strength. These morphofunctional traits are indicative of more active, predatory lifestyles in the three clades [58–61], but further suggest that gorgonopsians may have engaged in unique behaviors with their forelimbs (e.g., grappling large prey similar to more recent carnivores [62]). Therocephalians and gorgonopsians both have average

clade-wise postural scores converging on reptiles, but several taxa (*Gorgonops*, *Olivierosuchus*) have higher scores closer to therians (S8 Fig). This convergence may reflect more parasagittal postures in these taxa, but the range of forelimb poses would have been constrained by the therapsid shoulder girdle, with its caudolaterally facing glenoid [5].

Following theriodonts, further morphofunctional transformations to the forelimb associated with increasingly parasagittal postures have been proposed for cynodonts [5,13]. Cynodonts do shift closer to therians on the transitional landscape, but this movement is accompanied by considerable variation (Figs 1, 4 and S8) and evolutionary heterogeneity (Figs 5 and S9). Whereas postural trait variation in therapsids is phylogenetically structured by clade, in cynodonts it is more widespread across the phylogeny, implying enhanced evolutionary lability in posture and forelimb use. For functional traits, cynodonts generally optimize either humerus length, or a combination of strength and muscle force leverages (Figs 2, 3, S6, and S7), possibly reflecting adaptations to distinct, specialized ecologies (e.g., digging [63]). Most cynodonts are not optimized for humeral torsion, but those that are (e.g., *Lumkuiia*, *Riograndia*, *Brasilodon*) have been previously reconstructed with sprawling postures [64–66]. Likewise, lower torsion might indicate less sprawled postures in other taxa [67]. Postural scores in cynodonts are mainly in-line with modern reptiles, but some taxa achieve higher, more therian postural scores (*Massetognathus*, *Probainognathus*) (Figs 5 and S8). Given the disparity in cynodont forelimb functional and postural traits, it is likely that different species employed different postures, but were still constrained by the shoulder girdle to non-parasagittal (i.e., sprawling to semi-sprawling) limb poses [13,67–69]. Cynodont forelimb disparity is further explored by mammaliaforms, which possess disparate postural scores and functional traits: humerus length in *Megazostrodon*; length and torsion in *Morganucodon*; and length, 'spin' leverage and strength in *Borealestes* (Figs 2 and S7). The functional, and likely postural, diversity of mammaliaform humeri reflects the great ecomorphological disparity present in this group [70].

Our comprehensive survey of NMS found evidence of therian traits evolving multiple times, but limited support for fully parasagittal postures. Previous hypotheses place the origin of parasagittal posture in crown Mammalia [12] or even Theria [13]. Our two stem therians (*Adalatherium* and *Gobiconodon*) both plot with crown therians in morphospace (Figs 1 and S3) and on the transitional landscape (Figs 4 and 5). However, when we examine humerus traits, we find *Gobiconodon* strongly optimizes humeral torsion, which would preclude it from adopting a habitually parasagittal posture [71] (Figs 2 and S7). *Adalatherium,* on the other hand, possesses a functionally therian humerus, optimizing length, 'swing' force leverage and, to a lesser extent, inertial properties. Further support for therian parasagittal posture in *Adalatherium* comes from other aspects of the forelimb skeleton: the scapula has a ventrally facing glenoid facet, the coracoid portion of the glenoid fossa faces caudo-ventrally not laterally, and the ulna has a well-developed trochlear notch which likely restricted elbow motion to a single plane [72]. A similar condition in multituberculates, in combination with low humeral torsion, has been interpreted as evidence these taxa were also parasagittal [73,74]. Therefore, our results support the origins of fully parasagittal posture within stem therians [13], making it a late innovation in the grand scheme of synapsid evolution. Additionally, due to uncertainty in the phylogenetic arrangement of gondwanatherians (e.g., *Adalatherium*), eutriconodonts (e.g., *Gobiconodon*) and multituberculates [75,76], it is possible that adaptations for more parasagittal postures arose multiple times independently along the therian stem.

From evolutionary theory, we predicted that throughout their complex evolutionary journey, NMS were following an optimal pathway through morphospace (Hypothesis 4). Following prior work on synapsid evolution [5,12,13], we assumed that this optimal pathway would correspond to the acquisition of increasingly therian traits and postures. Instead, our Pareto optimality analyses clearly illustrate synapsid evolution as a series of successive adaptive radiations (Figs 6 and S10), with taxa repeatedly evolving away from the reconstructed optimal pathway. Pelycosaurs represent the first radiation of synapsids, but they do not diversify the forelimbs to the same extent as later synapsid grades [16]. The second radiation at the origin of Therapsida corresponds to a proposed morphofunctional shift in synapsid evolution based on important morphological transformations like the change in the scapular glenoid from a screw-shape to a more mobile hemi-sellar shape [5,12]. The removal of morphofunctional constraints on the forelimb potentially facilitated therapsid diversification, as they explored novel regions

of morphospace [16] and experimented with novel forelimb functions [52,61,77] (Figs 1–3 and S7). Some clades diversified in areas of high Pareto optimality (e.g., anomodonts), but biarmosuchians and theriodont therapsids explored Pareto suboptimal regions of morphospace and combinations of functional traits closer to those seen in therians and reptiles. This variability gives Therapsids a lower average Pareto optimality than pelycosaurs, providing strong supporting evidence for reduced evolutionary constraints [25]. Recovering theriodont therapsids as Pareto suboptimal also demonstrates that becoming increasingly therian was not the 'optimal' evolutionary pathway for synapsids until much later in their history.

Contrary to therapsids, cynodonts and mammaliaforms evolve on a multi-peak Pareto landscape, with optima corresponding to more robust and gracile humeri associated with the ancestral Cynodontia and derived Prozostrodontia landscapes, respectively (Figs 6 and S10). However, only a few taxa occupy either of these optimal regions, with most cynodonts and mammaliaforms occupying Pareto suboptimal regions between the peaks. We propose that the two peaks, representing distinct humerus morphologies and functional traits, constitute two ends of an ecological spectrum (e.g., fossorial to scansorial [64,65]), with more generalist taxa falling in the center due to conflicting selective pressures [24]. The gracile peak on the prozostrodont Pareto landscape is closer to the therian optimum and forms a Pareto optimal evolutionary pathway toward Theria. However, this is not the only optimal path on this landscape, as moving back toward the ancestral cynodont peak is also Pareto optimal. These multiple optima may have facilitated the disparate evolution of mammaliaforms [70], and the opposing evolutionary trajectories of monotremes and therians. Monotremes convergently re-evolve toward the robust, ancestral cynodont optimum, and therians realize a new set of humerus morphologies and functions. Accompanied by modifications to other aspects of the therian forelimb skeleton along the therian stem [72]—a mobile scapula, ventrally facing glenoid, and hinge-like elbow joint—this new morphology presumably removed the final barriers to achieving a habitually parasagittal forelimb posture [15].

## Conclusions

The evolution of parasagittal posture in Synapsida involved a fundamental reorganization of the musculoskeletal system and expansion of forelimb functional disparity, culminating in extant Theria. Our comprehensive examination of the 'sprawling-parasagittal' transition uses the framework of evolutionary adaptive landscapes to quantify differences in humerus morphology and functional traits between sprawling and parasagittal taxa and reconstructs the evolution of form and function in NMS. Our results indicate that although synapsids were ancestrally sprawling, they were both morphologically and functionally distinct from extant sprawling taxa. From this unique sprawling ancestral condition, we find that synapsid humerus morphofunctional evolution did not follow a single, optimum path to reach the parasagittal condition of modern therians. Some synapsid taxa stayed close to reconstructed group optima, but others diversified through morphospace. More gracile humeri with unique combinations of reptilian and therian traits evolved independently in biarmosuchian, gorgonopsian, and therocephalian therapsids, before an extended period of functional experimentation and variation throughout non-mammalian cynodont evolution. We reject previous hypotheses on the synapsid 'sprawling-parasagittal' transition as a series of discrete postural shifts, and instead support the view of synapsid evolution as a series of successive radiations, with major clades exhibiting considerable functional (and postural) variation. Our data on humerus morphology and functional trait evolution suggest that parasagittal posture evolved late, within stem therians. More sophisticated biomechanical modeling is required to resolve whole limb posture in specific taxa, but the present study, with its large sample of both fossil synapsids and extant comparators, provides a far more holistic picture of the 'sprawling-parasagittal' transition, and the assembly of the therian forelimb, than previous works considering only small numbers of exemplar taxa.

## Methods

### Sample and phylogeny

Humeri from a range of fossil synapsids (9 pelycosaur, 32 therapsid, 22 cynodont, and 2 stem therian specimens, representing at least 58 distinct taxa), were compared to a diverse array of extant taxa (5 salamanders, 56 reptiles, 3

monotremes, and 77 therian mammals, S1 Table). With respect to understanding the evolution NMS, monotremes possess several functional and morphological similarities to fossil taxa [14,36,78], and extant sprawling reptiles have long been used as analogues for NMS and other fossil amniotes [12,13,20]. Our extant sample covers a great diversity in body size and ecology, but all engage in quadrupedal terrestrial locomotion; volant and fully aquatic taxa were excluded because these derived specializations are not relevant to the 'sprawling-parasagittal' transition. We also included a small sample of generalized early fossil amniotes and temnospondyls ($n = 5$) to assist with establishing character polarity for the earliest NMS taxa. All information regarding data provenance, acquisition method, and processing to derive a 3D surface mesh is listed in S1 Table.

Relationships and branch lengths among extant species are based on consensus phylogenies downloaded from timetree.org [79], constructed based on comparison of all published phylogenies of the taxa of interest. Relationships of fossils (including fossil crown-mammals) are based on a combination of two existing large-scale phylogenies of living and extinct synapsids [80,81], which provide a backbone phylogeny of all non-mammalian synapsid fossils. Specimens missing from these two phylogenies were assigned the position of their closest relative present in the trees. The fossil supertree was time-calibrated with occurrence data downloaded from the Paleobiology Database (palaeobiodb.org), using the function *timePaleoPhy* from the *paleotree* package in R. The final phylogeny was then constructed by grafting the phylogenies of extant taxa onto the appropriate places in the fossil tree (e.g., salamanders as sister to *Eryops*, reptiles as sister to *Captorhinus*, etc.), and pruning the tree to only include sampled fossil taxa.

## Geometric morphometrics and morphospace

Humerus shape was quantified using a novel semi-automated approach for slice-based landmarking (S2 Fig). Three-dimensional surface meshes of each humerus were manually aligned to a global coordinate system using a standard protocol in Materialise 3-Matic (v14, Materialise, Leuven, Belgium). The long-axis of the humerus was aligned to the global Z-axis, and the bone was then rotated about its long-axis until the distal medial-lateral epicondyles were horizontal, parallel with the global *X* axis, and the extensor surface of the humerus (i.e., the face of the humerus to which the elbow extensor muscles attach), faced upwards (S4 Fig). The final sample consisted of only left humeri; any right humeri were mirrored. Once all humeri were consistently oriented, the 3D meshes were converted to '.ply' format and imported into R for landmarking.

Meshes were landmarked with a slice-based approach, using custom code based on the *Morphomap* R package [82] (S2 Fig). This slice-based landmarking produces a regular array of landmarks on the bone, describing the entire shape in detail, analogous to other morphometric methods, e.g., eigensurface analysis [83]. It works best on topologically simple structures with a well-defined long axis, such as limb bones. Each humerus was 'sliced' at 21 points evenly spaced along its length, by drawing a plane normal to the long axis that intersected with the bone mesh. In each of the 21 slices, the intersection between the plane and the mesh defined the bone's outer contour, and equidistant landmarks ($n = 24$) were placed around the contour's length. This produced a final set of 504 regularly spaced landmarks for each humerus, and these final landmark sets were subjected to general Procrustes alignment in the R package *geomorph* [84]. Differences in shape between groups, as well as potential allometric effects, were tested using a Procrustes ANOVA in the *geomorph* package [85] (S2 Table).

Data were ordinated using standard principal components analysis (PCA), which produced generally good separation between taxonomic and postural groups (S11 Fig). However, as we were specifically testing hypotheses of shape change as it relates to postural and functional change, we chose to incorporate a priori information on posture and ordinated the data using a bgPCA, in the R package *Morpho* [86]. This produced a morphospace that emphasized differences between postural groups (sprawling versus parasagittal versus 'unknown' for fossil taxa), and the bgPC axes are highly correlated with measured functional traits (Fig 2). Comparison with the traditional PCA shows a similar overall distribution of groups in morphospace, and that our bgPCA did not create 'false groups' [87]. Therefore, we proceeded with the more structured bgPCA morphospace in subsequent analyses.

### Measuring functional traits

**Functional humerus length.** Humerus length is an important component of locomotor performance (e.g., relative speed), as it is a major contributor to overall limb length, and relative humerus length positively correlates with lengths of other limb elements [88–92] (S12 Fig). Longer limbs produce longer strides and greater overall displacement [34,40,93,94]. Although there is considerable ecological variability in humeral length [7], we predicted that humeral length would increase throughout synapsid evolution. Functional humerus length—the length between the proximal and distal articular surfaces—was measured in 3-Matic as the distance from the proximal-most part of the humeral head, and the distal-most part of the radio-ulnar condyles (S4 Fig). After normalizing length to humeral centroid size, this differentiated long, gracile humeri from shorter, more robust humeri.

**Radius of gyration.** Limb bone shape is an important determinant of limb segment dimensions and mass properties, which determine the limb's rotational inertia, or how much torque, and hence effort, is required to move the limb during locomotion [95–98]. The inertial properties of the humerus directly affect those of the upper arm segment, which is generally the most massive forelimb segment in extant animals and so has a significant impact on total forelimb inertia [97–100] (S13 Fig). Rotational inertia of a limb segment is proportional to its mass ($m$) and to the square of the radius of gyration, or the distance from the center of mass to the center of rotation ($r$). Whereas segment mass is determined by size, radius of gyration is controlled by limb shape [96]; a shorter radius of gyration and more proximally positioned center of mass results in lower inertia and less energy needed to swing the limb. Radius of gyration was measured for each humerus model in 3-Matic (S4 Fig). Center of mass position for each humerus model was calculated automatically in 3-Matic, assuming the bone is solid, and the center of rotation was determined by fitting primitives (spheres, cylinders, or convex hulls) to the articular surface of the humeral head. To account for mass distribution along the proximodistal axis, radius of gyration was divided by humeral length [99,100]. We then took the inverse so that lower values corresponded to higher performance.

**Humeral torsion.** The presence of humeral torsion—an angular offset between the proximal and distal articulations—has been hypothesized to increase effective stride length in sprawling taxa, by placing the manus further forward during walking, proportional to the sine of the torsion angle [35]. Given its association with sprawling locomotion, we predicted that humeral torsion would decrease throughout synapsid evolution [101]. We measured humeral torsion in 3-Matic by fitting two planes along the humerus—one aligned with the humeral head passing through the greater and lesser tubercles (or the homologous attachment points of the subscapularis and supracoracoideus muscles), and one aligned with the ulnar and radial condyles (S4 Fig). The angle between these two planes was then taken as the metric of humeral torsion and transformed by taking the sine of the angle.

**Muscle leverage.** The anatomical and geometric arrangement of forelimb muscles are important determinants of functional performance. Muscle lever arms provide useful functional correlates for muscle action, as they directly correlate with muscle force and torque produced [102,103]. Of the muscle attachment sites easily identifiable across our humerus sample, the deltopectoral crest was the most reliable, and the deltoid and pectoral muscles that attach to it are both large muscles with important locomotor functions [15,41,45,78]. Therefore, we focused on the deltopectoral crest in our measurements of muscle leverage. Muscle in-levers were measured in 3-Matic from the most extreme point of the deltopectoral crest to the center of rotation at the humeral head (see the definition of rotational inertia), and then decomposed into separate components about different anatomical axes (S4 Fig).

For the $X$ and $Y$ axes, corresponding to humeral rotation in a vertical and horizontal plane, we calculated mechanical force advantages by dividing the muscle in-levers by humeral length (the out-lever for both these axes). We also measured velocity advantage by taking the inverse of force mechanical advantage (see note below). In both force and velocity advantages, the $X$ and $Y$ axis values were found to tightly correlate with one another, so these were averaged to produce single force and velocity advantage values. As humeral length is not the out-lever for long-axis rotation, and so cannot be

used to derive mechanical advantage, in-levers for the corresponding $Z$ axis were size normalized by dividing by humerus centroid size. This left us with three size-normalized metrics of muscle leverage: 'swing' force leverage (combining the $X$ and $Y$ axis force advantages) to move the limb through a vertical or horizontal arc with high torque; 'swing' speed leverage (combining the $X$ and $Y$ axis velocity advantages) to move the limb through a vertical or horizontal arc with high speed; and 'spin' leverage ($Z$ axis) to rotate the humerus about its long axis with high torque.

Adaptive landscape analyses often include traits that trade-off strongly with one another [19,23–25], but rarely are they exact inverses. While using separate surfaces for force and velocity may present methodological issues when only a few traits are modeled, it is less of a concern when considering multiple traits as we do in the present study. Further, it makes biological sense to include them both as individual traits that can be optimized as part of a combinatorial adaptive landscape, and we anticipated that some species would trade-off force for speed [15].

**Bending strength.** Limbs are subjected to varying loads during locomotion, and humeral strength determines the maximum external load that the bones can withstand. Although strength of the humerus is affected by both internal and external geometry, it has been shown that external geometry alone can provide a good approximation of whole bone mechanical properties, especially across broad phylogenetic samples that vary widely in gross morphology [104–106]. Additionally, limitations of how fossil specimens could be digitized (surface scanning and photogrammetry rather than computed tomography) necessitated restricting the analysis to surface structures. Therefore, our analysis of bending strength only relies on external geometry. When meshes were imported into R, the modified *Morphomap* code calculated mechanically relevant cross-sectional properties of each landmarked slice, effectively assuming that the interior of the bone was entirely solid. While this represents an imperfect assumption, we are confident that our methods are sufficient to compare across our broad, disparate sample of external humeral shapes.

For our strength metric, we chose the second moment of area about the x-axis, $I_x$, using our common coordinate system (S4 Fig). Second moments of area are commonly used metrics for bending resistance in bones [107,108] and are influenced by cross-sectional shape and external dimensions more so than by cross-sectional thickness [104–106,109], making them less sensitive to our assumptions about internal bone geometry. $I_x$ is indicative of resistance to dorsoventral bending, and at the specimen level correlated closely with other metrics such as resistance to anteroposterior bending ($I_y$) and torsional resistance ($J$). $I_x$ values were averaged across all landmarked slices for each specimen to incorporate information from the whole bone, and avoid uncertainty in consistently identifying the mid-diaphysis in such a diverse sample. Second moment of area has units in dimensions of length to the fourth power, so to normalize for humerus size we took the fourth root of our specimen average $I_x$ values before dividing by centroid size.

## Functional traits and performance surfaces

Performance surfaces were created using the R package *Morphoscape* [22]. For each of the seven functional traits measured, values were standardized by scaling them to range between 0 and 1, with 0 representing the lowest measured performance, and 1 representing the highest measured performance. Functional traits measured on humerus models from all specimens were mapped onto the morphospace using ordinary Kriging to interpolate trait values across the space. This resulted in seven unique performance surfaces, which clearly show high-level performance gradients and form-function relationships (Fig 2) but are more 'rugose' than those from other studies [18,19,22]. Kriging as an interpolation method tends to produce more rugose surfaces than others (e.g., polynomial surface fitting), but it also makes no a priori assumptions about the number of performance peaks or the shape of the performance surface. It should also be noted that while the landmarking protocol is free from biological homology, the functional measurements (e.g., torsion and muscle leverage) are not, which may cause some occasional disconnects between form and function visible as local hills and valleys in the performance surface (see S14 Fig).

## Adaptive landscapes

Adaptive landscapes were created using the R package *Morphoscape*. For each major taxonomic group of interest and for individual species, landscapes were calculated by iteratively summing the seven performance surfaces, where each one had its contribution to overall performance weighted. All possible combinations of weights were tested, ranging from 0 to 1 in increments of 0.05, which for seven functional traits resulted in a total of 230,230 possible adaptive landscapes. Combinations of weights were favored that maximized the height of the point of interest—a group mean, an individual taxon or a reconstructed ancestral nod—on the resulting landscape. For some taxa (e.g., reptiles and therians), despite clear functional differences between them, the very top landscapes tended to have similar combinations of weights; therefore, all final landscapes presented here are based on the mean of the top 10% of weight combinations. Differences in trait weighting combinations that produced optimal landscapes between taxonomic groups, and possible correlations of trait weighting with size, were statistically tested using an RRPP MANOVA in R (S3 Table).

## Postural and transitional landscapes

To trace the evolution of the humerus as synapsids transitioned from sprawling to parasagittal limb postures, we first calculated an extant 'sprawling' versus 'parasagittal' landscape. As therians were the only group in our analysis with parasagittal posture, the parasagittal landscape is equivalent to the therian adaptive landscape. However, the sprawling groups differed significantly from each other and were more broadly distributed across morphospace. Therefore, we created a composite sprawling landscape by overlaying the adaptive landscapes of individual sprawling groups ('herptiles', monotremes and non-synapsid fossils) and choosing the highest value in each grid cell as the value for the new landscape. High values on this new, composite landscape therefore indicated high functional performance for some form of sprawling, whether that be sprawling like a monotreme, reptile, salamander, or other tetrapod [13,20,36,45]. To calculate a transitional landscape and determine relative performance for sprawling or parasagittal postures in synapsids and other tetrapods, the difference between these 'sprawling' and 'parasagittal' landscapes was calculated by subtracting the values in each grid cell on the 'sprawling' landscape from the 'parasagittal' landscape. The resulting transitional landscape varies around 0, with negative values indicating increased performance on a sprawling landscape, and positive values indicating increased performance on the parasagittal therian landscape. Statistical differences in transitional landscape scores between groups and across body sizes were tested using an RRPP ANOVA in R (S5 Table).

## Phylogenetic modeling

To formally evaluate possible evolutionary shifts in posture and limb function across the 'sprawling-parasagittal' transition, we used the *SURFACE* package in R [32] to fit OU models for different evolutionary regimes for posture (scores on the transitional landscape) and limb function (trait weightings for a species adaptive landscape). As SURFACE can be prone to favoring overly complex models [110], we applied an AIC threshold of 4 during the model fitting process, so that more complex models with additional regimes had to provide a substantial improvement in model fit to be accepted over simpler models [111]. While SURFACE searches for shifts with no a priori assumptions, we also tested several explicit Brownian motion (BM) and Ornstein–Uhlenbeck (OU) models of postural evolution using mvMORPH [33]. We compared single-rate BM, BM with a trend, and single-peak OU models, as well as multi-rate BM and multi-peak OU models. For the multi-rate and multi-peak models, groupings were defined based on proposed timings of shifts in synapsid posture from Jenkins (shifts at Cynodontia and Theria) [13], Romer (shifts at Therapsida and Mammalia) [12] and Kemp (shifts at Therapsida and Mammaliaformes) [5]. We tested these against additional models where each synapsid grade was assigned a separate regime, or where further regimes were added for specific subclades (S5 Fig). For a full list of models tested, see S4 and S6 Tables.

Phylogenetic comparative methods can be sensitive to phylogenetic uncertainty and estimated branch lengths. Although there may be some uncertainty in the phylogenetic position of individual taxa or certain sub-clades [81], the higher-level

relationships of Synapsida are well recognized and established in the literature [6,57,112,113], and our analyses mainly focus on these high-level relationships, i.e., differences between grades. We assessed the effects of uncertainty in branch length by generating a distribution of 50 trees with variations in time scaling (*timepaleophy* function in *paleotree* with *dateTreatment = "minMax"*) and re-running the mvMORPH analyses to obtain a distribution of model support values. Analyzing posture (scores on the transitional landscape) revealed overwhelming support across all sampled trees for the original preferred model: an OU model with peaks for individual synapsid subclades (S4 Table) (S15 Fig). When analyzing limb function (trait weightings for a species adaptive landscape), we found a similar result to the main analysis (S6 Table): high support for models with regimes for individual synapsid subclades (both multi-rate BM and multi-peak OU) (S15 Fig). This sensitivity testing suggests that the major results of our comparative analyses are not strongly affected by uncertainties in branch length.

## Pareto landscapes

To determine whether synapsids evolved along optimal evolutionary trajectories through morphospace, maintaining maximum overall performance even if the performance traits themselves might change, we constructed a series of Pareto landscapes [25,46,114]. The grid cells in morphospace, which represent humerus morphologies, can be considered as solutions to an optimization problem, and they experience trade-offs between different performance metrics and fitness on different adaptive landscapes. A subset of these solutions will be Pareto optimal, i.e., no other solution has better or equal performance in all metrics [47]. This optimal subset of solutions is assigned rank 1, and then removed from the list of solutions before a second optimal subset (assigned rank 2) is determined. This process continues until all solutions have been ranked, thereby generating a Pareto ranking system.

Each Pareto landscape was calculated based on combining two adaptive landscapes reconstructed for major nodes in the Synapsid phylogeny—Synapsida, Therapsida, Cynodontia, Prozostrodontia and Theria. We ranked all locations in morphospace, based on their performance on the adaptive landscape reconstructed for a major synapsid node, and that for the next more derived major node (e.g., Synapsida and Therapsida, Therapsida and Cynodontia, etc.) toward which they might presumably be evolving. Optimality was ranked using a modified Goldberg ranking system [25,115], first calculating the optimal ranking ($R_O$) and then ranking again with the optimality of each metric reversed (suboptimal ranking, $R_S$), using the R package *Rpref* [116]. We then used the following equation to calculate Pareto optimality:

$$R_i = \frac{R_{si} - 1}{R_{Oi} + R_{Si} - 2}$$

This equation produces a linear rank from 0 to 1, where 1 denotes Pareto optimal regions and 0 denotes the most suboptimal regions. We plotted the Pareto landscapes for four regions of the Synapsid phylogeny, based on differences in the adaptive landscapes reconstructed along the synapsid backbone (Fig 2C). We also mapped Pareto optimality calculated for each specific subset of non-mammalian synapsid taxa onto the phylogeny—tip values were taken from the Pareto landscape, and ancestral states were calculated via maximum likelihood. Relationships of Pareto optimality with taxonomic group and centroid size were statistically tested in R using an RRPP ANOVA (S7 Table).

## Supporting information

**S1 Table. Specimen Information, including measurements needed to replicate the analysis (functional traits, and postural and taxonomic categories).** Also includes provenance and institutional details of each specimen in the dataset. (XLSX)

**S2 Table. Procrustes ANOVA model testing the effects of centroid size and taxonomic group on humerus shape, with additional pairwise comparisons between groups.** (XLSX)

 

**S3 Table. MANOVA model testing the effects of centroid size and taxonomic group on trait weighting optimization, with additional pairwise comparisons between groups.**
(XLSX)

**S4 Table. AIC scores and likelihoods for all evolutionary models of functional trait weighting tested using mvMORPH.**
(XLSX)

**S5 Table. ANOVA model testing the effects of centroid size and taxonomic group on transitional landscape sprawling-parasagittal scores, with additional pairwise comparisons between groups.**
(XLSX)

**S6 Table. AIC scores and likelihoods for all evolutionary models of transitional landscape sprawling-parasagittal scores tested using mvMORPH.**
(XLSX)

**S7 Table. ANOVA model testing the effects of centroid size and taxonomic group on Pareto optimality scores, with additional pairwise comparisons between groups.**
(XLSX)

**S8 Table. Acknowledgments for curatorial and collections staff.**
(XLSX)

**S1 Fig. Time-scaled supertree of all taxa included in this analysis.** Taxa are color-coded by group. The data underlying this figure can be found in S1 Table and S1 Data.
(PDF)

**S2 Fig. Illustration of the slice-based landmarking workflow on the humerus of a green iguana (*Iguana iguana*).** The top panels show a single plane slicing the mesh, and landmarks are placed around the contour resulting from the mesh-plane intersection. This is then repeated at intervals along the length of the bone to place landmarks across the entire surface.
(PDF)

**S3 Fig. Labeled morphospace of all taxa included in this analysis.** Taxa are listed and numbered in alphabetical order. Taxa are color-coded by group. The data underlying this figure can be found in S1 Table and S1 Data.
(PDF)

**S4 Fig. Functional measurements taken in 3-Matic, illustrated on the humerus of a tree monitor (*Varanus prasinus*).** Prior to measurement, all humeri were aligned to a common coordinate system. All linear measurements were taken in mm, angular measurements were taken in degrees. Humerus length, distance from the center of mass to the center of rotation, in-lever for the muscles attaching to the deltopectoral crest (broken down in x, y, and z components) and torsion angle between the proximal and distal ends of the humerus.
(PDF)

**S5 Fig. Evolutionary regimes mapped onto our phylogeny to illustrate the *a priori* evolutionary hypotheses tested using mvMORPH.** These are the hypotheses of synapsid forelimb transformation proposed by Jenkins, Romer, and Kemp. We also tested each major synapsid grade as its own regime, as well as adding sub-regimes for specific groups. The data underlying this figure can be found in S1 Table and S1 Data.
(PDF)

**S6 Fig. PCA (top) and SURFACE analysis (bottom) of optimal landscape trait weights calculated for individual taxa.** SURFACE analysis indicates shifts in the multi-variate evolutionary regime of adaptive landscape trait weights across synapsid evolution. Some convergent regimes were detected, and are color-coded the same. The data underlying this figure can be found in S1 Table and S1 Data.
(PDF)

**S7 Fig. Trait weightings for landscapes calculated on individual fossil synapsid taxa.** The data underlying this figure can be found in S1 Table and S1 Data.
(PDF)

**S8 Fig. Ridgeline (top) and phenogram (bottom) showing calculated scores on the transitional 'sprawling-parasagittal' landscape for different non-mammalian synapsids.** Colored lines on the phenogram show the median, upper, and lower quartile values for extant therians, herptiles, and monotremes. Dashed line on the phenogram indicates the synapsid phylogeny backbone. The data underlying this figure can be found in S1 Table and S1 Data.
(PDF)

**S9 Fig. SURFACE analysis indicating shifts in the univariate evolutionary regime of sprawling-parasagittal landscape scores.** No convergent regimes were detected. The data underlying this figure can be found in S1 Table and S1 Data.
(PDF)

**S10 Fig. Ridgeline plot (top) and phenogram (bottom) showing calculated Pareto Optimality for different non-mammalian synapsids.** Dashed line on the phenogram shows the backbone of synapsid phylogeny. The data underlying this figure can be found in S1 Table and S1 Data.
(PDF)

**S11 Fig. Standard principal components analysis (PCA) morphospace of all taxa, for comparison with the between-groups principal component analysis (bgPCA) in** Figs 1 **and** S3. The data underlying this figure can be found in S1 Table and S1 Data.
(PDF)

**S12 Fig. Relationships between relative humerus length and relative forearm (radius or ulna) length across extant tetrapods.** There is a consistent, significant positive relationship between longer humeri and longer forearm elements, demonstrating that humerus morphology encodes useful information about the forelimb as a whole. The data underlying this figure can be found in S1 Data.
(PDF)

**S13 Fig. Mass (top) and inertia (bottom) properties for forelimb segments taken from the literature, illustrating the relatively higher mass and inertia of the upper arm—made up of the humerus and surrounding soft tissue—relative to the forearm.** Mammal data from Coatham and colleagues (2021), reptile data from Mcaulay and colleagues (2023). The data underlying this figure can be found in S1 Data.
(PDF)

**S14 Fig. Raw data for the performance surfaces—functional trait data plotted onto the morphospace—before interpolation through Kriging.** Larger, more opaque points correspond to higher trait values. The data underlying this figure can be found in S1 Data.
(PDF)

**S15 Fig. Analyzing sensitivity of evolutionary model fits to branch length uncertainty.** Phylogeny showing ranges of nodal ages across 50 trees (A). AIC weights for each of the models of trait (B) and posture (C) evolution tested using mvMORPH (S5 Fig), across a distribution of 50 trees. Favored models match the preferred models recovered by the main analysis and discussed in the main text (see Results and S4 and S6 Tables). The data underlying this figure can be found in S1 Table and S1 Data.
(PDF)

**S1 Data. Landmark coordinates for all humerus specimens, supertree of all taxa in our dataset, and R code which, in combination with the information in S1 Table, can be used to replicate the analysis and recreate the main figures and results, and S1, S3, S5–S11 and S14 Figs. We also include the R code used for the slice-based landmarking procedure, literature values for limb proportions and inertial properties needed to replicate S12 and S13 Figs, as well as the sample of 50 trees for the branch-length sensitivity analysis in S15 Fig.**
(7Z)

## Acknowledgments

For assistance with specimen scanning and data acquisition, we thank Jacqueline Lungmus, Spencer Hellert, Mark Wright, Peter Bishop, Blake Dickson, John Nyakatura, Tom Kemp, Roger Benson, and Elsa Panciroli. We thank Blake Dickson for assistance with *Morphoscape*, and Daniel Rhoda for helpful discussions about adaptive landscape analyses. We are grateful to collections managers and curators from institutions all over the world who facilitated access to specimens—either physically or digitally—without whom this research would not have been possible (See S8 Table).

## Author contributions

**Conceptualization:** Robert J. Brocklehurst, Stephanie E. Pierce.

**Formal analysis:** Robert J. Brocklehurst, Magdalen Mercado.

**Funding acquisition:** Kenneth D. Angielczyk, Stephanie E. Pierce.

**Investigation:** Robert J. Brocklehurst, Magdalen Mercado.

**Methodology:** Robert J. Brocklehurst, Magdalen Mercado.

**Project administration:** Kenneth D. Angielczyk, Stephanie E. Pierce.

**Resources:** Kenneth D. Angielczyk, Stephanie E. Pierce.

**Software:** Robert J. Brocklehurst.

**Supervision:** Robert J. Brocklehurst, Kenneth D. Angielczyk, Stephanie E. Pierce.

**Visualization:** Robert J. Brocklehurst.

**Writing – original draft:** Robert J. Brocklehurst, Stephanie E. Pierce.

**Writing – review & editing:** Robert J. Brocklehurst, Magdalen Mercado, Kenneth D. Angielczyk, Stephanie E. Pierce.

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
