## [Editor Report · Decision Letter 0]

Dear Dr Brocklehurst, 

Thank you for submitting your manuscript entitled "Adaptive landscapes unveil the complex evolutionary path to mammalian forelimb function and posture" for consideration as a Research Article by PLOS Biology.

Your manuscript has now been evaluated by the PLOS Biology editorial staff and I'm writing to let you know that we would like to send your submission out for external peer review.

Once your full submission is complete, your paper will undergo a series of checks in preparation for peer review. After your manuscript has passed the checks it will be sent out for review. To provide the metadata for your submission, please Login to Editorial Manager (https://www.editorialmanager.com/pbiology) within two working days, i.e. by May 17 2024 11:59PM.

Kind regards,

Rolai Roberts

Roland Roberts, PhD

Senior Editor

PLOS Biology

rroberts@plos.org

---

## [Decision Letter · Decision Letter 1]

Dear Dr Brocklehurst,

Thank you for your patience while your manuscript "Adaptive landscapes unveil the complex evolutionary path to mammalian forelimb function and posture" was peer-reviewed by PLOS Biology. Your work was assessed and discussed by the PLOS Biology editorial team and by four independent reviewers. Based on the reviews, which you will find at the end of this email, I regret that we will not be pursuing your manuscript for publication at PLOS Biology. Please accept my apologies for the delay incurred while we sought advice from an Academic Editor during the challenging summer months.

You'll see that reviewer #1 is very positive, but asks you to consider several issues (fossil preservation, body size correlation). Reviewer #2 is less impressed, explaining why s/he thinks that it’s incremental and descriptive, and needs more rigorous stats. Reviewer #3 is somewhat positive, but remains sceptical, questioning your reliance on extant monotremes, and wondering how valid your morphological proxies for function are, suggesting that you should test their approach on extant mammals. Reviewer #4 seems to think that s/he is reviewing for a very specialised journal, and wants you to spell out your limitations more, especially the focus on the humerus, and to improve how you consider and present your biomechanical parameters.

Overall, the reviewers raise a number of non-overlapping concerns, including sparse sampling and limited novelty, and during cross-commenting, several reviewers explicitly said that it was better suited to a more specialised journal (I note that one of the reviewers mentions Journal of Mammalian Evolution).

I am sorry that we cannot be more positive on this occasion and hope the reviewer reports will help you in preparing your manuscript for submission elsewhere.

While we cannot consider your manuscript further for publication in PLOS Biology, we would suggest transferring your manuscript, with reviews, to PLOS ONE instead (http://journals.plos.org/plosone/). 

PLOS ONE is a peer-reviewed journal that accepts original research that contributes to the base of academic knowledge. The review process at PLOS ONE focuses on scientific validity, strong methodology and high ethical standards, and the journal's inclusive scope and broad reach means that research published in PLOS ONE will be read, cited and used by researchers across many disciplines. In this case, the PLOS ONE Academic Editors will also take the feedback received from the reviewers at PLOS Biology into account when reaching a decision, which should increase the efficiency of the review process. Please note that the journals are editorially independent and we therefore cannot guarantee the outcome if you choose to pursue a transfer. 

If you would like to submit your work to PLOS ONE, please click the following link:

<DeepLinkData><DeepLinkTypeID>27</DeepLinkTypeID><peopleID>2049383</peopleID><userSecurityID>046c307f-8722-4c2e-9f66-d048a7dab3a4</userSecurityID><documentID>59078</documentID><revision>1</revision><manuscriptNumber>PBIOLOGY-D-24-01322</manuscriptNumber><docSecurityID>0834bfcd-10fe-4ce6-8717-199c64242bd3</docSecurityID></DeepLinkData>

If you do NOT wish to submit your work to PLOS ONE, please click this link to decline: 

<DeepLinkData><DeepLinkTypeID>28</DeepLinkTypeID><peopleID>2049383</peopleID><userSecurityID>046c307f-8722-4c2e-9f66-d048a7dab3a4</userSecurityID><documentID>59078</documentID><revision>1</revision><manuscriptNumber>PBIOLOGY-D-24-01322</manuscriptNumber><docSecurityID>0834bfcd-10fe-4ce6-8717-199c64242bd3</docSecurityID></DeepLinkData>

Should you choose to transfer your submission to PLOS ONE, you will receive a confirmation email within 24-48 hours of accepting the transfer. Your submission details and manuscript files will be transferred automatically; however, because all PLOS journals vary in submission requirements, once in the PLOS ONE Editorial Manager site, you will be asked to provide additional information before you can finalize your new submission to PLOS ONE. If you have any questions, please feel free to contact the journal at plosone@plos.org.

I hope you understand the reasons for this decision and that the option of publishing your work in PLOS ONE might be useful. Thank you for your support of PLOS and of open-access publishing.

Sincerely,

Roli Roberts

Roland Roberts, PhD

Senior Editor

PLOS Biology

rroberts@plos.org

REVIEWERS' COMMENTS:

Reviewer #1: 

This manuscript uses innovative morphometrics and adaptive landscape modelling to interrogate the evolution of the sprawling-parasagittal transition in synapsids. This is clearly a very well written and complete manuscript and I think it should be considered for publication with only very minor revisions. 

I have one question about the sampling of fossils. It seems like this semi-automated pseudolandmarking approach would be sensitive to fossil preservation. What was the consideration of fossil specimen selection with regard to preservation/quality? Were any deformed or damaged fossils digitally reconstructed? There are only sparse details about repairing a single specimen in the supplemental table. 

My only other concern is whether the different optima are body-size correlated. This is discussed briefly on lines 288-293 but I think it could be spelled out a bit more clearly for non-specialist readers. For example, is there a correlation between Sprawling-Parasagittal Score and estimated body mass? What about Pareto Optimality and estimated body mass? 

Do you want to make a prediction as to what the unique posture/gait of pelycosaurs might have meant for biology/ecology/behaviour(see line 279)- as you do with Therocephalians on line 304-305?

Line 362 space needed in "humeriassociated"

Reviewer #2:

This is a very clearly written but dense manuscript that seeks to add to our knowledge of the tempo and mode of morphological innovations/changes associated with the so called 'sprawling-parasagittal transition' in vertebrate evolution. Here the researchers focus on the humerus - or rather 3D shape analyses of it.

This study makes the point that what was previously known about this issue is gleaned from just one or a few representatives for critical groups (i.e. a very sparse sample of the diversity). This seems to have led previous researchers to propose a stepwise view of postural change. This study increases the sampling somewhat and finds that this stepwise view is not supported in this sample.

The approach here in terms of shape analyses is not new - they have been used by the authors before in various papers. The shape analyses are not controversial in that there is no doubt that they characterize shape in a way that can be studies comparatively. Although I wonder if such a highly engineered description of 'shape' is necessary or desirable - for example how does selection act on specifically these complex numerical description of shape. This is something that the authors also, even if unknowingly, worry about as they often seek to link their numerical descriptors of shape back to the raw morphology. Anyhow this is a minor point and arguably just a matter of preference rather than any sort of critical problem with what the authors have done.

It is true that the authors have increased the sampling from previous work, and this looks numerically impressive. However, when viewed in terms of the sampling across the total number of species in the groups studied is still very space. So, how can we be sure that further sampling will not change our knowledge again?

I think the manuscript is slightly underwhelming in terms of the testing among the hypotheses proposed and the phylogenetic work is very basic. The results are largely descriptive and there are no real statistical tests among the hypotheses. Thus, there is only really narrative evidence for one hypothesis over the others. I think a better statistical formulation of the expectations of the hypotheses might be needed to truly show some decisive evidence.

I don't mean to be too critical of this work, I think it is well done and clearly written. However, I think there are limits to what can be said from these data and analytical approaches. I think this work does represent an increase in our knowledge in the area, but I think it is incremental rather than transformative. With this then, and the necessary density of the writing needed to explain the study, I think this manuscript would be far more suitable for a more specialized journal where I think the readership will appreciate the contribution better.

Reviewer #3:

I have now read the manuscript entitled 'Adaptive landscapes unveil the complex evolutionary path to mammalian forelimb function and posture' submitted to PLOS Biology. In the manuscript the authors use analyses of humerus shape in an extensive sample of fossil taxa to infer function and posture. They then use these data to establish adaptive landscapes and then infer pareto optimality criteria to test the idea that synapids followed an 'optimal' path to the erect parasagittal gate in extant therians. Although there are many thing I really like about this paper including the sample of extinct taxa and the general approach which I found original and exciting, there are also a number of elements that made me question the inferences drawn from the data. I think the overall conclusions make sense and are sound but I wonder whether the data actually support these conclusions. I also found that the authors at time are a bit overly eager to replace shape (what they studied) with function which unfortunately remains unmeasured and only inferred. Below I detail what I mean by this. I hope my comments and suggestions are useful to the authors and apologise if I have missed information in the manuscript or supplementary information that could have changed my evaluation of the manuscript.

1) sample

Although the authors have an amazing and beautiful sample of extinct taxa I found the sample of extant taxa rather limited and potentially biased. For example, extant monotremes are heavily used as a proxy for the ancestor of early mammals but how relevant are they really? In the discussion the authors themselves highlight this and point out that they are in reality highly derived forms. 

2) function 

The authors talk about function, performance and posture throughout the manuscript but never do they actually measure these things. All they do is infer proxies thereof based on analyses of shape and morphology. Thus I would strongly urge the authors to be precise in the introduction, results and the discussion of the manuscript and use appropriate language. You did not measure any of these but only used proxies thereof. Moreover, I have some doubts about the relevance of these functional proxies: 

limb length: you state that humerus is an important contributor to limb length, yet I doubt that it contributes to functional limb length in sprawling animals. In these animals the humerus is oriented more in the horizontal plane and as such does not really contribute to functional limb length. As posture changes humerus length will contribute more to functional limb length but the problem is you infer posture partly from length making this rather circular.

rotational inertia: using volume is very delicate as a proxy for mass as this is dependent on cortical thickness. I doubt you had information on this parameter but maybe I missed it. If you did include variation in cortical thickness in calculating the rotational inertia then please state this explicitly in the methods.

Muscle leverage: I was a bit surprised by the rather broad conclusions derived from using a single muscle proxy, the dectopectoral crest. I would argue that this may be a good proxy for humeral retraction but inferring anything about swing from the position of this crest seems a bit tricky. Moreover, the functionally relevant trait to measure is muscle moment arm. Yet, muscle moment arms change throughout the movement and the movement is unknown... so how relevant/useful is this proxy really ?

Bending strength: the same limitation here as for rotational inertia, without information on cortical thickness any estimation of bending strength becomes extremely difficult to interpret. 

Overall I was wondering how good these functional proxies really are in estimating function, performance or posture... In reality I would have liked to have see a data set on extant mammals where the authors test the validity of these proxies the way they have been calculated before applying them to fossil taxa. If they could provide this and show that they are in fact decent proxies then I would be convinced by the approach. As it stands, everything that follows (calculation of adaptive landscapes etc... is derived from these proxies and thus the overall interpretation as well. This seems like a major limitation of the study.

minor comments:

line 37: what do we really know about the limb muscles in these extinct forms. Or is this inferred from the comparison between reptiles and mammals ?

line 47: seems like a word is missing after 'presumed'

line 52: I agree the humerus is interesting and may provide interesting insights, without the scapula our ability for inferring posture seems extremely limited. The scapula is the primary articulation between the forelimb and the body, not the humerus. As such I would argue that the scapula would be an even better structure to look at. I understand that humeri are probably better preserved in the fossil record, but be honest about why you chose that structure.

line 57: you did not measure functional performance so don't write that. Same on line 70 and further in the manuscript.

line 82: your landmarks are not homology free... you use a procrustes superimposition so your landmarks become at least spatially homologous. 

line 88: extant monotremes re highly derived so how useful are they in the overall discussion ?

line 99: you can only infer posture, what you measured is humerus morphology, be precise in your language.

line 113: you did not quantify function, you only infer it ! just say function was inferred or estimated.

line 141: I would not say salamanders and lizard (your herptiles) are characterized by fast limb movements ... rather they have slow humerus movement. This illustrates that you have to be careful in the interpretation of your functional proxies. Here your limited taxon sampling for extant species may bias the data set as well.

Lines 154 and further: I was lacking information on how important size is as a driver of your functional proxies... you come back to this in the discussion while saying that behaviors like burrowing or large size/mass may be important drivers of humerus morphology but I would have like to see these caveats addressed earlier in the manuscript.

Discussion: I found the discussion rather long and not always focused on the important elements. I would have like to see the limitations of the data and the functional and postural inferences discussed in greater detail before embarking upon length discussions on evolutionary transitions. 

line 392: rather than more sophisticated biomechanical models I think we first need data on extant taxa to validate your proxies.

Reviewer #4:

[see also the attached PDF]

Using performance landscapes, the authors explore the evolution of the locomotor system (as inferred by humeral morphology) for non-mammalian synapsids in attempt to document changes in limb posture among mammals and their more ancient kin. This is an interesting study, and I think that it is worthy of publication in Journal of Mammalian Evolution. However, there are some points that need to be addressed prior to publication.

One of the main concerns about this study is that the authors need to devote more of the discussion to the limitations of their study. Their study is indeed informative, but the humerus is only one component of the locomotor system, or even just forelimb anatomy. Therefore, due to other factors, such as muscle anatomy, the anatomy of other limb bones/segments, etc, could it be possible that the taxa that were found to have locomotor postural differences in this study could have in fact had practically similar locomotor styles? In other words, could there be many-to-one mapping of humeral morphology to a given mode of locomotion or posture? This is not something requires a change of the study design or a reanalysis, but it needs to be specifically addressed in the Discussion.

The authors also need to be clear and consistent in their terminology. For instance, the authors specify there is muscle force leverage, muscle speed leverage, and muscle spin leverage. However, at times throughout the text, the authors simply refer to muscle leverage, and it is not clear which of the three is meant or if somehow all three are meant simultaneously. Also, the names of some variables/parameters seem to change between the text and the figures. The authors need to carefully check this for consistency.

The authors need to more carefully consider their biomechanical parameters, or at the very least how they present their parameters. For instance, a second moment of area has to be measured about a specified axis, which the authors don't mention. Likewise, a rotational inertia is also always about a reference axis, not simply a point. However, if the authors were sure to measure each bone in a fixed orientation, then using a simple point could represent an axis coming 90° out of a plane of orientation. So, they need to specify the bone's orientation. Also, the authors might strongly consider trying to measure a parameter that measures resistance to long-axis torsional loads. The parameter that comes to mind in this respect the polar moment of area. The polar moment of area is ideally applied to circular cross sections, but it might still be somewhat informative for the authors' study, given the role that torsional loads play for sprawling taxa (as indicated by the work of Blob & Biewener).

Kind of going back to my point about many-to-one mapping, I would say that the authors really do not need to discuss the other limb segments when it comes to two parameters in particular: limb length and rotational inertia. Given that the length and mass proportions can greatly vary among limb segments in relation to locomotor modes (at least in mammals and maybe birds' hindlimbs), the authors should mention whether "optimized" humeral length or rotational inertia, as recovered by their study, is sufficient enough to give some indication of overall the overall forelimb's morphology and function, or could there be some ambiguity based upon how length, mass, and (consequently) inertia varies among limb segments. One thing the authors could try is to *qualitatively* report in the Discussion how, say, radius length or olecranon length varies relative to the humerus among taxa clustering in the humerus-based adaptive landscape. Of course, these bones or all their features might not be known for fossil taxa, but this doesn't have to be done in extreme detail. Just enough to give the reader some insight.

When it comes to weightings of their biomechanical parameters, the authors should also give some indication in the text (not just figures) whether the optimizing of a parameter in the landscape entails high or low values of that parameter. This is because, from biomechanical principles, a short humerus might be "optimum" for a given locomotor mode (e.g., a more cursorial taxon), whereas a long humerus might be "optimum" for others (e.g., a more scansorial taxon). You could say the same about rotational inertia. So, I think it's insufficient to simply say a trait is optimized-it's vague from a biomechanical perspective. Please be more specific about this point in the main text.

The use of the term "herptiles" is for me really jarring. Initially, I didn't mind it, but by the end of the manuscript, somehow it became taxing (this is subjective preference, I wholly admit). I think the authors are better off defining (for the purpose of their study) that "reptiles" is taken to mean "non-avian reptiles." As I said though, this point is wholly subjective and my preference.

Other more minor comments are highlighted in the attached PDF.

---

## [Editor Report · Decision Letter 2]

Dear Dr Brocklehurst,

Thanks for your patience while we considered your Appeal of our previous decision to Reject your manuscript "Adaptive landscapes unveil the complex evolutionary path to mammalian forelimb function and posture." As mentioned in my early email, we discussed the points that you raised with an Academic Editor; for transparency, this Academic Editor was not involved in the previous decision, so is a "fresh pair of eyes."

In light of the Academic Editor's input, we would like to invite you to revise the work to thoroughly address the reviewers' reports. In case it proves helpful, I've included (with permission) a lightly edited version of the Academic Editor's advice to us; these comments relate directly to points made by the reviewers and to your objections/arguments. The AE is being somewhat modest; they are well-placed to make the big-picture assessment, but are simply deferring on more detailed issues. My reading of their comments is broadly that while your rebuttal was understandably adversarial, you should seek the common ground with the reviewers and try to appreciate what they were "really" after, with a view to strengthening the paper. I'm sure this will be easier to do in the knowledge that we are now giving you the opportunity to revise and resubmit...!

AE'S ADVICE (edited):

My general impression was that both, reviewer 2 (the most critical of them all) and the authors themselves, might be both missing each other’s points. I found myself agreeing with several points made by reviewer 2 but I should also say that this might also be because we come a traditions/areas of expertise more similar to the reviewer than to the authors, hence my own prejudices might be similar to his. That said I actually thought (keep in mind that I am not an expert on the subject) that the paper might eventually be nice contribution if the authors take the revision seriously, including those from reviewer 2... I do understand their [the authors'] point though, but I hope they are able to see some of the criticism more positively because it might indeed increase the quality of the paper.

Below I list some points that I thought were most of the “tension” was, including the authors' reluctance, in my opinion, to see the point of the criticism.

1) “The suggestion by Reviewer #2 that our work is not quantitative because we did not perform explicit statistical tests is patently false. Our work is HIGHLY quantitative, every figure is made from quantitative data and modeled using complex code that took many years to develop.” Sure, it is quantitative in that sense, but that was not what I think the reviewer was going after, the reviewer’s focus was, I think on the statistical aspect of hypothesis testing, once you have described the adaptive landscape in a highly quantitative matter as the authors have in fact pointed out they did. I am not sure how to implement this here (I am skating on thin ice…) but I wonder if the authors could investigate the individual description for each species on the transitional ‘sprawling-parasagittal’ landscape, their adaptive score (height), beyond ancestral state reconstruction and SURFACE analysis (see lines 175 to 176), to for example, more explicitly test different models of trait evolution that map into their narrative (e.g. test if such landscape occupation is gradual, if different clades indeed have different optima, etc.). That said, it is also fair to say that the analysis behind figure 6, do to some extent, represent explorations in a quantitative way, although not explicitly including hypothesis testing based on contrasting different statistical models… Not sure this makes much sense, but I felt that reviewer 2 was going after some kind of explicit hypothesis, comparing different statistical models that represent different evolutionary scenarios more explicitly, and that could indeed make the paper better.

2) (very similar to the point above) “In the Discussion we interpret our quantitative data within the full scope of 300+ million years of synapsid evolution; while this requires qualitative correlations, such correlations are entirely appropriate for macroevolutionary studies.” If I got it right, reviewer 2 refers to the point that “results are largely descriptive and there are no real statistical tests among the hypotheses. Thus, there is only really narrative evidence for one hypothesis over the others”. I guess the point here is that some large-scale studies in macroevolution these days might indeed use a toolbox of sophisticated statistical methods, that might go beyond what was implemented here. I do sympathize with all the effort put by the authors; they have done a great job in assembling the dataset and quantitatively describing the morpho-space/adaptive landscape. It is quite impressive indeed, but I guess the reviewer’s might be that they can do better, and perhaps that a general audience would expect a more hypothesis driven/statistical oriented paper.

3) “The comment from Reviewer #2 that our phylogenetic work is “very basic” further misses the point of our manuscript. Adaptive landscapes are not the same as phylogenetic comparative methods (PCMs), as they allow a fully multivariate analysis of traits and interpretations of direct form-function trade-offs, not possible with PCMs (Polly et al 2020). If adaptive landscapes are not the reviewer’s “preferred method”, that is not a reason for rejection.” This might indeed regard the analysis in the PCM sense (if so my comments would be the same as the two above) but I wonder if the reviewer’s point was not in fact focused on the phylogenetic reconstruction itself (described on the methods section of the paper). The phylogenetic reconstruction does seem a bit too simple, but I am not sure if it would be possible to do something better. It might…

4) “Further, the reviewer [in this case reviewer 3] suggests we validate our proxies on extant mammals; we included a large sample of extant mammals, reptiles and salamanders (n = 138) in our paper for the exact purpose of validating the use of these proxies in inferring posture and locomotion! We point the editor to the beginning of the Discussion where we explicitly detail which traits in the extant taxa correlate with posture.” I might be missing something here, but again I suspect the reviewer and authors are alluding to different things, or they would do things differently. In my mind, and I could be wrong here since form-function studies are really not my area of expertise, the reviewer point is that before adding those extant in the analysis (which will certainly help the analysis), one should test, using only extant taxa where we have data on form, and more importantly, on function (in this case posture I presume), that the morphological proxies and the methods used indeed point to evidence that those proxies are good proxies. I might be wrong, but I suspect that the reviewer is particularly concerned by how the analysis shown in figure 4, 5 and 6 does really translate into a proper description of the form-function relationship and how the methodology underlying those figures can be applied to extinct taxa before we validate it to extant taxa. That said, I am not sure how well established those relationships are (maybe they are trivial and hence the authors are not really seeing the point here). I do not know the topic and this literature at depth. Based on this sentence here in the results “As expected, therians had the highest scores and monotremes had the lowest, indicating trait combinations most and least associated with parasagittal postures respectively”, it feels that the authors consider their own results to justify the methods. Hence, in one hand it seems that the reviewer is asking for direct test for method validation using extant taxa only, and in the other the authors suppose those relationships are well established for living taxa and hence this is not an issue.

[END OF AE'S COMMENTS]

Given the extent of revision needed, we cannot make a decision about publication until we have seen the revised manuscript and your response to the reviewers' comments. Your revised manuscript is likely to be sent for further evaluation by all or a subset of the reviewers.

**IMPORTANT - SUBMITTING YOUR REVISION**

*Re-submission Checklist*

*Published Peer Review*

*PLOS Data Policy*

*Blot and Gel Data Policy*

Sincerely,

Roli Roberts

Roland Roberts, PhD

Senior Editor

PLOS Biology

rroberts@plos.org

REVIEWERS' COMMENTS:

Reviewer #1:

This manuscript uses innovative morphometrics and adaptive landscape modelling to interrogate the evolution of the sprawling-parasagittal transition in synapsids. This is clearly a very well written and complete manuscript and I think it should be considered for publication with only very minor revisions.

I have one question about the sampling of fossils. It seems like this semi-automated pseudolandmarking approach would be sensitive to fossil preservation. What was the consideration of fossil specimen selection with regard to preservation/quality? Were any deformed or damaged fossils digitally reconstructed? There are only sparse details about repairing a single specimen in the supplemental table.

My only other concern is whether the different optima are body-size correlated. This is discussed briefly on lines 288-293 but I think it could be spelled out a bit more clearly for non-specialist readers. For example, is there a correlation between Sprawling-Parasagittal Score and estimated body mass? What about Pareto Optimality and estimated body mass?

Do you want to make a prediction as to what the unique posture/gait of pelycosaurs might have meant for biology/ecology/behaviour(see line 279)- as you do with Therocephalians on line 304-305?

Line 362 space needed in "humeriassociated"

Reviewer #2:

This is a very clearly written but dense manuscript that seeks to add to our knowledge of the tempo and mode of morphological innovations/changes associated with the so called 'sprawling-parasagittal transition' in vertebrate evolution. Here the researchers focus on the humerus - or rather 3D shape analyses of it.

This study makes the point that what was previously known about this issue is gleaned from just one or a few representatives for critical groups (i.e. a very sparse sample of the diversity). This seems to have led previous researchers to propose a stepwise view of postural change. This study increases the sampling somewhat and finds that this stepwise view is not supported in this sample.

The approach here in terms of shape analyses is not new - they have been used by the authors before in various papers. The shape analyses are not controversial in that there is no doubt that they characterize shape in a way that can be studies comparatively. Although I wonder if such a highly engineered description of 'shape' is necessary or desirable - for example how does selection act on specifically these complex numerical description of shape. This is something that the authors also, even if unknowingly, worry about as they often seek to link their numerical descriptors of shape back to the raw morphology. Anyhow this is a minor point and arguably just a matter of preference rather than any sort of critical problem with what the authors have done.

It is true that the authors have increased the sampling from previous work, and this looks numerically impressive. However, when viewed in terms of the sampling across the total number of species in the groups studied is still very space. So, how can we be sure that further sampling will not change our knowledge again?

I think the manuscript is slightly underwhelming in terms of the testing among the hypotheses proposed and the phylogenetic work is very basic. The results are largely descriptive and there are no real statistical tests among the hypotheses. Thus, there is only really narrative evidence for one hypothesis over the others. I think a better statistical formulation of the expectations of the hypotheses might be needed to truly show some decisive evidence.

I don't mean to be too critical of this work, I think it is well done and clearly written. However, I think there are limits to what can be said from these data and analytical approaches. I think this work does represent an increase in our knowledge in the area, but I think it is incremental rather than transformative. With this then, and the necessary density of the writing needed to explain the study, I think this manuscript would be far more suitable for a more specialized journal where I think the readership will appreciate the contribution better.

Reviewer #3:

I have now read the manuscript entitled 'Adaptive landscapes unveil the complex evolutionary path to mammalian forelimb function and posture' submitted to PLOS Biology. In the manuscript the authors use analyses of humerus shape in an extensive sample of fossil taxa to infer function and posture. They then use these data to establish adaptive landscapes and then infer pareto optimality criteria to test the idea that synapids followed an 'optimal' path to the erect parasagittal gate in extant therians. Although there are many thing I really like about this paper including the sample of extinct taxa and the general approach which I found original and exciting, there are also a number of elements that made me question the inferences drawn from the data. I think the overall conclusions make sense and are sound but I wonder whether the data actually support these conclusions. I also found that the authors at time are a bit overly eager to replace shape (what they studied) with function which unfortunately remains unmeasured and only inferred. Below I detail what I mean by this. I hope my comments and suggestions are useful to the authors and apologise if I have missed information in the manuscript or supplementary information that could have changed my evaluation of the manuscript.

1) sample

Although the authors have an amazing and beautiful sample of extinct taxa I found the sample of extant taxa rather limited and potentially biased. For example, extant monotremes are heavily used as a proxy for the ancestor of early mammals but how relevant are they really? In the discussion the authors themselves highlight this and point out that they are in reality highly derived forms.

2) function

The authors talk about function, performance and posture throughout the manuscript but never do they actually measure these things. All they do is infer proxies thereof based on analyses of shape and morphology. Thus I would strongly urge the authors to be precise in the introduction, results and the discussion of the manuscript and use appropriate language. You did not measure any of these but only used proxies thereof. Moreover, I have some doubts about the relevance of these functional proxies:

limb length: you state that humerus is an important contributor to limb length, yet I doubt that it contributes to functional limb length in sprawling animals. In these animals the humerus is oriented more in the horizontal plane and as such does not really contribute to functional limb length. As posture changes humerus length will contribute more to functional limb length but the problem is you infer posture partly from length making this rather circular.

rotational inertia: using volume is very delicate as a proxy for mass as this is dependent on cortical thickness. I doubt you had information on this parameter but maybe I missed it. If you did include variation in cortical thickness in calculating the rotational inertia then please state this explicitly in the methods.

Muscle leverage: I was a bit surprised by the rather broad conclusions derived from using a single muscle proxy, the dectopectoral crest. I would argue that this may be a good proxy for humeral retraction but inferring anything about swing from the position of this crest seems a bit tricky. Moreover, the functionally relevant trait to measure is muscle moment arm. Yet, muscle moment arms change throughout the movement and the movement is unknown... so how relevant/useful is this proxy really ?

Bending strength: the same limitation here as for rotational inertia, without information on cortical thickness any estimation of bending strength becomes extremely difficult to interpret.

Overall I was wondering how good these functional proxies really are in estimating function, performance or posture... In reality I would have liked to have see a data set on extant mammals where the authors test the validity of these proxies the way they have been calculated before applying them to fossil taxa. If they could provide this and show that they are in fact decent proxies then I would be convinced by the approach. As it stands, everything that follows (calculation of adaptive landscapes etc... is derived from these proxies and thus the overall interpretation as well. This seems like a major limitation of the study.

minor comments:

line 37: what do we really know about the limb muscles in these extinct forms. Or is this inferred from the comparison between reptiles and mammals ?

line 47: seems like a word is missing after 'presumed'

line 52: I agree the humerus is interesting and may provide interesting insights, without the scapula our ability for inferring posture seems extremely limited. The scapula is the primary articulation between the forelimb and the body, not the humerus. As such I would argue that the scapula would be an even better structure to look at. I understand that humeri are probably better preserved in the fossil record, but be honest about why you chose that structure.

line 57: you did not measure functional performance so don't write that. Same on line 70 and further in the manuscript.

line 82: your landmarks are not homology free... you use a procrustes superimposition so your landmarks become at least spatially homologous.

line 88: extant monotremes re highly derived so how useful are they in the overall discussion ?

line 99: you can only infer posture, what you measured is humerus morphology, be precise in your language.

line 113: you did not quantify function, you only infer it ! just say function was inferred or estimated.

line 141: I would not say salamanders and lizard (your herptiles) are characterized by fast limb movements ... rather they have slow humerus movement. This illustrates that you have to be careful in the interpretation of your functional proxies. Here your limited taxon sampling for extant species may bias the data set as well.

Lines 154 and further: I was lacking information on how important size is as a driver of your functional proxies... you come back to this in the discussion while saying that behaviors like burrowing or large size/mass may be important drivers of humerus morphology but I would have like to see these caveats addressed earlier in the manuscript.

Discussion: I found the discussion rather long and not always focused on the important elements. I would have like to see the limitations of the data and the functional and postural inferences discussed in greater detail before embarking upon length discussions on evolutionary transitions.

line 392: rather than more sophisticated biomechanical models I think we first need data on extant taxa to validate your proxies.

Reviewer #4:

[see also the attached PDF]

Using performance landscapes, the authors explore the evolution of the locomotor system (as inferred by humeral morphology) for non-mammalian synapsids in attempt to document changes in limb posture among mammals and their more ancient kin. This is an interesting study, and I think that it is worthy of publication in Journal of Mammalian Evolution. However, there are some points that need to be addressed prior to publication.

One of the main concerns about this study is that the authors need to devote more of the discussion to the limitations of their study. Their study is indeed informative, but the humerus is only one component of the locomotor system, or even just forelimb anatomy. Therefore, due to other factors, such as muscle anatomy, the anatomy of other limb bones/segments, etc, could it be possible that the taxa that were found to have locomotor postural differences in this study could have in fact had practically similar locomotor styles? In other words, could there be many-to-one mapping of humeral morphology to a given mode of locomotion or posture? This is not something requires a change of the study design or a reanalysis, but it needs to be specifically addressed in the Discussion.

The authors also need to be clear and consistent in their terminology. For instance, the authors specify there is muscle force leverage, muscle speed leverage, and muscle spin leverage. However, at times throughout the text, the authors simply refer to muscle leverage, and it is not clear which of the three is meant or if somehow all three are meant simultaneously. Also, the names of some variables/parameters seem to change between the text and the figures. The authors need to carefully check this for consistency.

The authors need to more carefully consider their biomechanical parameters, or at the very least how they present their parameters. For instance, a second moment of area has to be measured about a specified axis, which the authors don't mention. Likewise, a rotational inertia is also always about a reference axis, not simply a point. However, if the authors were sure to measure each bone in a fixed orientation, then using a simple point could represent an axis coming 90° out of a plane of orientation. So, they need to specify the bone's orientation. Also, the authors might strongly consider trying to measure a parameter that measures resistance to long-axis torsional loads. The parameter that comes to mind in this respect the polar moment of area. The polar moment of area is ideally applied to circular cross sections, but it might still be somewhat informative for the authors' study, given the role that torsional loads play for sprawling taxa (as indicated by the work of Blob & Biewener).

Kind of going back to my point about many-to-one mapping, I would say that the authors really do not need to discuss the other limb segments when it comes to two parameters in particular: limb length and rotational inertia. Given that the length and mass proportions can greatly vary among limb segments in relation to locomotor modes (at least in mammals and maybe birds' hindlimbs), the authors should mention whether "optimized" humeral length or rotational inertia, as recovered by their study, is sufficient enough to give some indication of overall the overall forelimb's morphology and function, or could there be some ambiguity based upon how length, mass, and (consequently) inertia varies among limb segments. One thing the authors could try is to *qualitatively* report in the Discussion how, say, radius length or olecranon length varies relative to the humerus among taxa clustering in the humerus-based adaptive landscape. Of course, these bones or all their features might not be known for fossil taxa, but this doesn't have to be done in extreme detail. Just enough to give the reader some insight.

When it comes to weightings of their biomechanical parameters, the authors should also give some indication in the text (not just figures) whether the optimizing of a parameter in the landscape entails high or low values of that parameter. This is because, from biomechanical principles, a short humerus might be "optimum" for a given locomotor mode (e.g., a more cursorial taxon), whereas a long humerus might be "optimum" for others (e.g., a more scansorial taxon). You could say the same about rotational inertia. So, I think it's insufficient to simply say a trait is optimized-it's vague from a biomechanical perspective. Please be more specific about this point in the main text.

The use of the term "herptiles" is for me really jarring. Initially, I didn't mind it, but by the end of the manuscript, somehow it became taxing (this is subjective preference, I wholly admit). I think the authors are better off defining (for the purpose of their study) that "reptiles" is taken to mean "non-avian reptiles." As I said though, this point is wholly subjective and my preference.

Other more minor comments are highlighted in the attached PDF.

---

## [Decision Letter · Decision Letter 3]

Dear Dr Brocklehurst,

Thank you for your patience while we considered your revised manuscript "Adaptive landscapes unveil the complex evolutionary path to mammalian forelimb function and posture" for publication as a Research Article at PLOS Biology. This revised version of your manuscript has been evaluated by the PLOS Biology editors, the Academic Editor, and by two of the original reviewers. Unusuaully, at the request of the Academic Editor, we also sought input from two new reviewers.

Based on the reviews and on our Academic Editor's assessment of your revision, we are likely to accept this manuscript for publication, provided you satisfactorily address the remaining points raised by the reviewers and the following data and other policy-related requests.

IMPORTANT - please attend to the following:

a) Please could you make your Title slightly more explicit for our broader readership? We suggest "Adaptive landscapes unveil the complex evolutionary path from sprawling to mammalian forelimb function and posture"

b) Please address the remaining concerns raised by the reviewers. IMPORTANT: The Academic Editor kindly provided some additional guidance as to how to address the continuing points made by reviewer #2; this advice can be found at the foot of this letter.

c) Please address my Data Policy requests below; specifically, we need you to supply the numerical values underlying Figs 1ABC, 2ABC, 3B, 4ABCD, 5, 6ABC, S1, S3, S5, S6, S8, S9, S10, S11, S12, S13, S14, either as a supplementary data file or as a permanent DOI’d deposition. I note that you already have a supplementary zipped folder, Data S1, but it is not clear whether the R files that it contains are sufficient to reproduce all of the Figs; please can you clarify and (if necessary) supply the required numerical values?

d) Please cite the location of the data clearly in all relevant main and supplementary Figure legends, e.g. “The data underlying this Figure can be found in S1 Data” or “The data underlying this Figure can be found in https://zenodo.org/records/XXXXXXXX

e) Please make any custom code available, either as a supplementary file or as part of your data deposition.

We expect to receive your revised manuscript within two weeks. 

*Published Peer Review History*

*Press*

Sincerely,

Roli Roberts

Roland Roberts, PhD

Senior Editor

rroberts@plos.org

PLOS Biology

DATA POLICY:

Regardless of the method selected, please ensure that you provide the individual numerical values that underlie the summary data displayed in the following figure panels as they are essential for readers to assess your analysis and to reproduce it: Figs 1ABC, 2ABC, 3B, 4ABCD, 5, 6ABC, S1, S3, S5, S6, S8, S9, S10, S11, S12, S13, S14. NOTE: the numerical data provided should include all replicates AND the way in which the plotted mean and errors were derived (it should not present only the mean/average values).

CODE POLICY

DATA NOT SHOWN?

REVIEWERS' COMMENTS:

Reviewer #2:

At the end of my last referee report, I noted that I find this work interesting and the data collection impressive. Furthermore, from my scrutiny, I believe the methods used in the manuscript have been applied correctly.

However, I would like to emphasize that my role as a referee is to provide my opinion based on the journal's criteria for publication. In this case, the authors do not appear to have taken my previous comments very seriously. Instead, their response comes across as quite combative. In light of their revision and response, I will restate my key points below:

1. Apologies for my sloppy use of "tempo and mode."

I appreciate the clarification provided by the authors on this point.

2. Shape vs Function

The authors use a homology-free method to characterize shape. I find this approach very interesting, but its relevance to evolutionary analysis is unclear. Shape and function are not synonymous — shape is the product of function alongside many other (often unknown) factors that trade off with it. The hypotheses tested in this manuscript are explicitly about function. Therefore, it would be far more appropriate to measure homologous functional traits directly, or to generate biomechanical traits that are clearly linked to locomotor changes. As it stands, it feels as though the authors have applied a novel method to capture shape, and then retrofitted the resulting data to test hypotheses about function — a conceptual mismatch that is still unresolved.

3. Sample Size

The increased sample size is an improvement, but the dataset remains sparse relative to the total possible number of species. The authors claim that expanding the sample further would not significantly alter the results — but they provide no justification for this statement. As presented, this is pure speculation.

4. Phylogenetic Tree and Comparative Methods

I appreciate the authors' clarification here, but several issues remain:

The Phylogeny: The authors employ state-of-the-art metatree methods to generate a total-group synapsid phylogeny including nearly 1,900 fossil and non-fossil taxa etc. This is indeed a reasonable approach given the data available, and I acknowledge that the authors are working with difficult material. However, this approach comes with major limitations. Most notably, the tree is presented with a degree of topological certainty that is simply not warranted. There is no real discussion of topological uncertainty, which is a significant omission.

Branch Lengths and Divergence Dates: These are notoriously difficult to estimate accurately, even with large molecular datasets. In a tree built this way, branch lengths are almost certainly poorly estimated (again, this is a known limitation, not a criticism of the authors' effort). However, this uncertainty could have a major impact on the results from comparative methods. There is no attempt to assess or account for this potential issue in the manuscript.

Comparative Methods: The authors use SURFACE to identify 'adaptive landscapes'. While this method is widely used, I would not describe it as particularly cutting-edge. More importantly, the reliability of any comparative method depends heavily on the quality of both the tree and the data. Given the phylogenetic uncertainties discussed above, the results of the SURFACE analysis could easily be compromised. Simply adding a couple of extra statistical tests does not address this underlying structural problem.

Conclusion

In summary, while I appreciate the authors' response and their effort to improve the manuscript, several of my fundamental concerns remain unresolved. I continue to feel that the study applies an interesting shape-analysis technique, but does so in a way that does not align well with the core functional hypotheses. Additionally, the phylogenetic uncertainties are not sufficiently acknowledged, despite their potential to undermine key conclusions.

Reviewer #3:

[identifies himself as Anthony Herrel]

The revised version of the manuscript entitled 'Adaptive landscapes unveil the complex evolutionary path to mammalian forelimb function and posture' appears much improved. The authors have done a lot of work on the manuscript and I think it can be published in its current form. I appreciate the detailed replies to the reviewer comments provided by the authors. For some functional proxies I am still not 100% sure that the in vivo data needed to validate hem really exist, but I think the authors make solid arguments for most. 

Reviewer #5:

Summary of Paper

This paper looks at the evolution of posture in non-mammalian synapsids (NMS) using the morphology of the humerus. The comprehensive dataset of NMS provides valuable new insight into the postural transition from NMS to mammals. I encourage publication, but do recommend some minor revisions.

Comments 

I noted that previous reviewers took issue with the taxonomic sample. I would argue this is a non-issue. The taxonomic sample is incredibly comprehensive for NMS and arguably the most impressive aspect of the paper. No reasonable addition of specimens could be made to this dataset.

The language change to "functional traits" when referring to functional proxies clarifies the fact that these are not direct measurements of function, and the references to other papers that test these proxies directly with function adequately address the issue of function not being measured directly

The new method for semi-automated landmarking is quite clever approach to structures that have the problem of few landmarks.

The explicit hypotheses laid out at the end of the introduction are well addressed throughout the results and discussion. The connection between the analyses and hypotheses were well explained.

My most significant recommendation would be to move away from kriging when creating the adaptive landscapes. Although kriging makes no a prior assumptions, it tends to overfit and produce unrealistically rugged topologies. In this paper, the landscapes for Radius of Gyration, Torsion, and Spin Lever seem to have particularly convoluted topologies (Figure 2A). Something like the second-order polynomial surface used by Dickson and Pierce (2019) would make more sense. See Rhoda and Angielczyk, "Multivariate functional adaptive landscapes and how we make them" in Friedman, Matt, and Jeffrey Wilson Mantilla. 12TH NORTH AMERICAN PALEONTOLOGICAL CONVENTION UNIVERSITY OF MICHIGAN 17-21 JUNE 2024. (2024).

Methods section Line 583-584 says adaptive landscapes were calculated for each major taxonomic group of interest, but Line 613 in the methods and 128 - 130 in the results say that adaptive landscapes were produced for each species in the dataset. Language should be clarified.

Add explained percentage of variation on the bgPCA axes (Figure 1 and S3)

Reviewer #6:

I am commenting on this revised manuscript after previous reviewer comments. I have a number of points for the authors to address to ensure justification and clarity of methods and interpretations.

Justification for the use of between-groups principal components analysis: I don't see a good reason to forceable separate taxonomic groups when the objective is to analyse functional morphology. I would say that solid justification for the use of bgPCA should be made here (above defending it against potential artefacts that it may generate).

Slice-based landmarking method: The landmarking method is described as 'novel' whereas it is essentially identical to that pioneered by Polly (2008, Mammalian Evolutionary Morphology, pp 167-196; Polly and MacLeod 2008, Palaeontologia Electronica, 11: 10A) called 'eigensurface analysis'. Acknowledgements of the history of this approach, and its limitations and strengths, should be made clear to the reader.

'Unique mode of sprawling': the Abstract suggests that when a group falls into a separate morphological area, it must also have a separate (or 'unique') functional form, essentially equating differences in morphology to differences in function. However, the authors later acknowledge the potential 'many-to-one' mapping that may mean that these different humeral morphologies may in fact have equivalent sprawling function.

COMMENTS FROM THE ACADEMIC EDITOR:

I think they can address the limitations posed by reviewer #2 in text. The only alternative would require a new full tree inference where topological uncertainty would be addressed and explicitly estimated to be later used. At this point I am not sure this is possible or desired given all the work already done here.... That said I think "a real discussion of topological uncertainty, which is a significant omission."(reviewer #2) should be done. I think we can condition the acceptance on them being really explicit on this

Also as pointed out by reviewer #6 :"The landmarking method is described as 'novel' whereas it is essentially identical to that pioneered by Polly (2008, Mammalian Evolutionary Morphology, pp 167-196; Polly and MacLeod 2008, Palaeontologia Electronica, 11: 10A) called 'eigensurface analysis'. Acknowledgements of the history of this approach, and its limitations and strengths, should be made clear to the reader."

I noticed that all (or most of) the reviewers emphasized the need to be more clear about the limitations of the different methods (each one a different aspect....). I wonder if it might be beneficial to include a small section on caveats which could even point out what could be done in the future to better take those into account.

---

## [Editor Report · Decision Letter 4]

Dear Robert,

Thank you for the submission of your revised Research Article "Adaptive landscapes unveil the complex evolutionary path from sprawling to upright forelimb function and posture in mammals" for publication in PLOS Biology. On behalf of my colleagues and the Academic Editor, Tiago Quental, I'm pleased to say that we can in principle accept your manuscript for publication, provided you address any remaining formatting and reporting issues. These will be detailed in an email you should receive within 2-3 business days from our colleagues in the journal operations team; no action is required from you until then. Please note that we will not be able to formally accept your manuscript and schedule it for publication until you have completed any requested changes.

IMPORTANT: I've asked my colleagues to include the following editorial request alongside their own: I note that in your data availability statement you say, "For replicating raw data collection, custom code for placing landmarks based on the existing R package Morphomap, will be made available from the corresponding author on request." We do not allow on-request statements, so please include this additional custom code in S1 Data, and change the data availability statement accordingly.

Sincerely,

Roli

Senior Editor

PLOS Biology

rroberts@plos.org